# A Problem-Oriented Perspective and Anchor Verification for Code Optimization

**Tong Ye[1], Tengfei Ma[2], Xuhong Zhang[1]***, **Hang Yu[3], Jianwei Yin[1], Wenhai Wang[1]***

[1]Zhejiang University, [2]Stony Brook University, [3]AntGroup
`tongye@zju.edu.cn`

## Abstract

Large Language Models (LLMs) have shown remarkable capabilities in solving various programming tasks, such as code generation. However, their potential for code optimization, particularly in performance enhancement, remains largely unexplored. This paper investigates the capabilities of LLMs in optimizing code for minimal execution time, addressing a critical gap in current research. The recently proposed code optimization methods construct program optimization pairs based on iterative submissions from the same programmer for the same problem. However, this approach confines LLMs to local performance improvements, neglecting global algorithmic innovation. To overcome this limitation, we adopt a completely different perspective by reconstructing the optimization pairs into a problem-oriented approach. This allows for the integration of various ideas from multiple programmers tackling the same problem. Furthermore, we observe that code optimization presents greater challenges compared to code generation, often accompanied by "optimization tax". Recognizing the inherent trade-offs in correctness and efficiency, we introduce a novel anchor verification framework to mitigate this "optimization tax". Ultimately, the problem oriented perspective combined with the anchor verification framework significantly enhances both the correct optimization ratio and speedup to new levels.

## 1 Introduction

LLMs and Code LLMs, such as GPT-4 Series (OpenAI et al., 2024), CodeLLaMA (Roziere et al., 2023), DeepSeek-Coder Series (Guo et al., 2024; Zhu et al., 2024) and Qwen-Coder Series (Yang et al., 2024; Hui et al., 2024), have demonstrated remarkable capabilities in software engineering tasks, garnering significant attention from both academia and industry. In tasks such as code completion and code generation, Code LLMs achieve high correctness rates (Pass@K) on widely used benchmarks like EvalPlus (Liu et al., 2023), LiveCodeBench (Jain et al., 2025), and Big-CodeBench (Zhuo et al., 2025). However, despite these advancements, the code produced by these LLMs often falls short in real-world applications. It may lack the necessary optimizations to meet specific performance and efficiency requirements (Shi et al., 2024; Niu et al., 2024). As a result, the generated code often requires further refinement and optimization to align with practical constraints.

While low-level optimizing compilers and performance engineering tools have made significant advancements (Alfred et al., 2007; Wang & O'Boyle, 2018), they primarily focus on hardware-centric optimizations. High-level performance considerations, such as algorithm selection and API usage, still rely heavily on manual intervention by programmers. Automating high-level code optimization remains a major challenge and has yet to be widely explored. Code optimization can be approached from various angles. In this work, we specifically focus on time performance, with an emphasis on minimizing program execution time, given its critical importance in practical applications.

In the field of code performance optimization, the construction of optimization pairs is a critical challenge. Unlike code generation, which only requires the collection of correct code, code performance optimization demands semantically equivalent code pairs with varying levels of efficiency. This

---

*Corresponding authors.

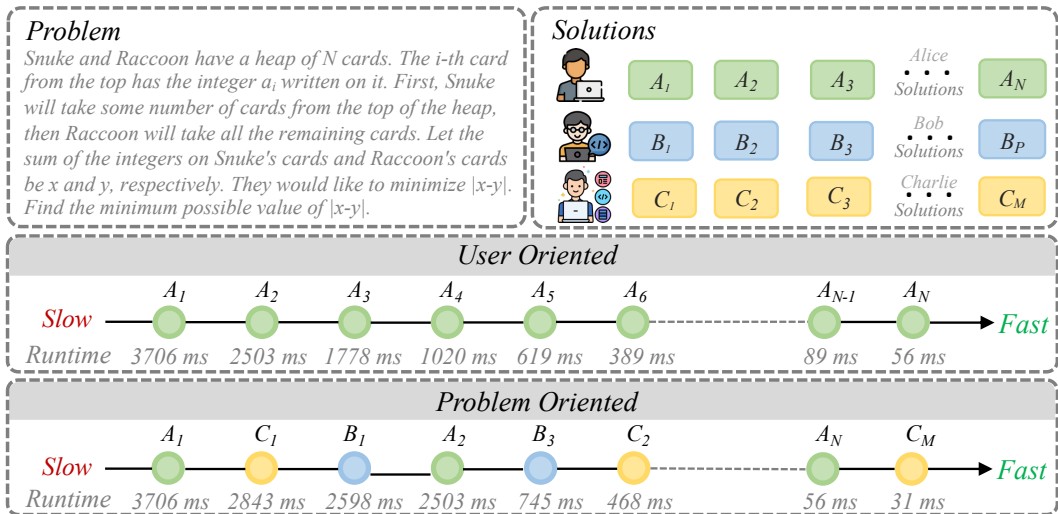

Figure 1: For a given problem, different users submit and iterate on their code solutions. The user oriented perspective constructs optimization pairs based on the individual users. In contrast, the problem oriented perspective analyzes all solutions for the problem to build trajectories.

dual requirement, ensuring both functional correctness and measurable performance improvements, makes dataset creation considerably more complex. Recent study (Shypula et al., 2024) partly addressed this challenge by collecting user iterative submissions from programming platforms, such as LeetCode, creating code optimization pairs (each consisting of less efficient code and its semantically equivalent, more efficient counterpart). By utilizing these optimization pairs, researchers have initially demonstrated the potential of LLMs in code optimization tasks through domain fine-tuning.

However, the current approach of constructing code optimization pairs from iterative submissions by the same user has significant limitations. We refer to this as the **User-Oriented** approach. As shown in Figure 1, a user initially submits a solution to a programming problem, but early versions may fail to meet the system's time constraints due to excessive computational overhead. Through iterative refinements, the user eventually arrives at a more efficient solution. This process captures the user's submission trajectory, which is used to construct optimization pairs such as $(A_1, A_2), (A_2, A_3), ..., (A_{N-1}, A_N)$. While this approach naturally reflects the direction of code optimization, it is inherently constrained by the thought patterns of a single programmer. Consequently, improvements tend to be incremental, building upon existing logic and paradigms. A substantial number of intuitive examples (Figure 16 - 19) in Appendix T also demonstrate this phenomenon. In contrast, real-world code optimization thrives on collaborative diversity. Code review and refactoring processes deliberately involve multiple programmers to overcome cognitive inertia, with innovation arising from the synthesis of diverse perspectives. Inspired by this insight, we hypothesize that combining different users' perspectives is beneficial for code optimization. Therefore, we propose shifting from the user-oriented perspective to a **Problem-Oriented** perspective. We restructure optimization pairs by incorporating solutions from multiple programmers addressing the same problem. As illustrated in the last part of Figure 1, solutions from different users, ordered by runtime, form a completely new optimization trajectory for the given problem. This problem-oriented perspective encourages a diverse range of innovative ideas, fostering a more holistic optimization process that better mirrors the complexity and creativity of program optimization. Multidimensional analysis and experimental results show that adapting Code LLMs to problem-oriented optimization pairs greatly enhances optimization capabilities, leading to significant improvements in both optimization ratios ($31.24\% \rightarrow 58.90\%$) and speedup ($2.95\times \rightarrow 5.22\times$).

Simultaneously, code optimization is essentially a dual-objective process. It aims to enhance code efficiency while ensuring the accuracy of the optimized code. However, in practice, we find that there is often a conflict between these two goals. That is, the code optimized by LLM can't be guaranteed to be completely correct. We refer to this phenomenon as the "optimization tax". To mitigate the practical challenge, we present an innovative anchor verification framework. The core idea is to leverage the "slow but correct" nature of pre-optimized code to enhance the accuracy of

the optimized code. Specifically, the anchor verification framework draws from a widely used test case execution feedback mechanism like Chen et al. (2023); Wei et al. (2024); Chen et al. (2024a) in code generation tasks. These methods depend on synthesized test cases and bidirectional execution filtering to verify test cases and code. However, anchor verifiaction framework differs from these methods. Instead of directly synthesizing complete test cases, it first uses the LLM to interpret the "slow code" and generate test case inputs. Then, it treats the "slow code" as a test case anchor to real execution to produce precise outputs for these test case inputs. By pairing each test case input with its corresponding execution output, we create complete and verified test cases. These verified test cases are then used for the iterative refinement of the "optimized code". Further experimental results show that anchor verification framework further unlocks performance bottlenecks and pushes code optimization to new levels, significantly improving both the optimization ratio ($58.90\% \rightarrow 71.06\%$), speedup ($5.22\times \rightarrow 6.08\times$), and correctness ($61.55\% \rightarrow 74.54\%$). In summary, the contributions are:

- To the best of our knowledge, we are the first to introduce a problem-oriented perspective for code optimization. This perspective not only enhances the richness and diversity of optimization pairs but also significantly alleviates the data scarcity issue in code optimization.
- We reveal the performance bottlenecks in code optimization, identify the "optimization tax" and introduce the anchor verification framework to effectively mitigate the bottlenecks. The anchor verification framework fully utilizes the characteristics of the code optimization task: the code to be optimized, though inefficient, is at least functional correct.
- Multi-dimensional analysis and experiment results validate the effectiveness and robustness of both the problem-oriented perspective and the anchor verification framework, significantly and simultaneously improving the optimization ratio, speedup, and correctness.

**Overall Architecture:** This paper begins by presenting a problem-oriented perspective for constructing optimization pairs in Section 2. In the associated experiments, we identify the "optimization tax" and highlight the relevant performance bottlenecks. Building on these findings, we propose a novel anchor verification framework in Section 3 to further unleash LLMs' optimization potential.

## 2 PROBLEM-ORIENTED CODE OPTIMIZATION

In this section, we first introduce the key distinctions of the user-oriented perspective and the problem-oriented perspective in § 2.1. Subsequently, we carry out in-depth multi-dimension analyses of both user-oriented and problem-oriented optimization pairs (§ 2.2). After that, we discuss the adaptation of Code LLMs to two perspective optimization pairs (§ 2.3) and furthermore conduct the optimization pairs percentage analysis and learning edit patterns analysis in § 2.4.

### 2.1 PROBLEM-ORIENTED OPTIMIZATION PAIRS

**User-Oriented Perspective.** In the current research, code optimization pairs are derived from PIE, introduced by Shypula et al. (2024), which focuses on optimizing program execution time by utilizing human programmers' submissions from a wide range of competitive programming tasks on CodeNet (Puri et al., 2021). A key aspect of developing PIE is recognizing the typical workflow of programmers: when faced with a problem, they usually begin with an initial solution and then iteratively refine it. As shown in Figure 1, for a given problem $\mathcal{P}$, users (*Alice, Bob, etc.*) have their submission trajectories, filter out incorrect submissions, and sort the rest in chronological order.

$$\text{Alice valid submissions: } [A_1, A_2, A_3, \ldots, A_N]$$
$$\text{Bob valid submissions: } [B_1, B_2, B_3, \ldots, B_P]$$
$$\text{Charlie valid submissions: } [C_1, C_2, C_3, \ldots, C_M]$$

The user-oriented optimization pairs are constructed by extracting sequential pairs from each user's submission trajectory. For example, Alice's valid submissions generate optimization pairs such as $(A_1, A_2), (A_2, A_3)$, and so on, while Charlie's valid submissions result in optimization pairs like $(C_1, C_2), (C_2, C_3)$, and so forth. Ultimately, aggregating all these optimization pairs forms the complete user-oriented optimization dataset (PIE).

**Problem-Oriented Perspective.** While user-oriented optimization pairs indicate the direction of optimization, as previously noted, they are inherently confined by the cognitive patterns of a single

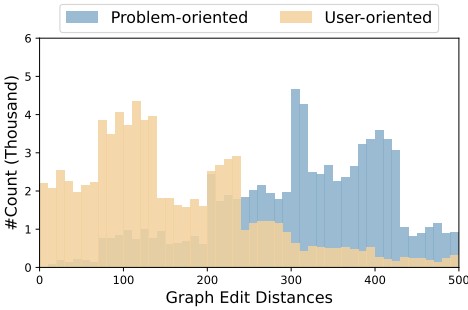

Figure 2: Program Structural Analysis of the Disparities between Problem-oriented Optimization Pairs and User-oriented Optimization Pairs using Graph Edit Distance (GED) metric.

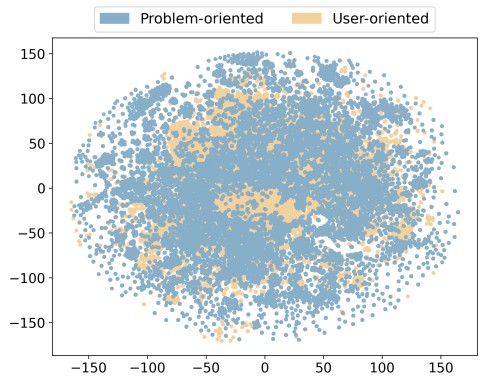

Figure 3: Semantic Representation Analysis of Problem-oriented and User-oriented Pairs.

programmer. The detailed instances in Appendix T also illustrate this point, intuitively showing that the overall problem-solving approach and logical framework remain largely unaltered. Therefore, we shift the perspective on optimization pairs and propose a problem-oriented construction method. Specifically, we regard all submissions for the same problem $\mathcal{P}$ from different users as a single group, thereby breaking down the barriers between different users. We sort all valid user submissions for the same $\mathcal{P}$ based on the marked runtime and map them onto the same optimization trajectories:

$$\text{All users for problem } \mathcal{P}: [A_1, C_1, B_1, A_2, B_3, C_2, \ldots, C_M]$$

Subsequently, we construct optimization pairs along the problem-oriented trajectory, such as $(A_1, C_1), (C_1, B_1), (C_1, B_2), etc.$ Ultimately, this simple but not trivial process yields the novel problem-oriented optimization dataset. This new perspective not only reflects the direction of optimization but also integrates the diverse strategies and algorithms of different programmers.

**Alleviating the Data Scarcity.** The problem-oriented perspective also offers a significant advantage in terms of scale. Let's assume there are $\mathcal{P}$ problems, each with $\mathcal{U}$ users, and each user has $n_u$ valid submissions. The user-oriented and problem-oriented perspectives exhibit a substantial divergence in the scaling of optimization pairs:

$$\text{\# optimization pairs of user oriented} = \frac{1}{2} \cdot \sum_{p=1}^{\mathcal{P}} \sum_{u=1}^{\mathcal{U}} C_{n_u}^2$$

$$\text{\# optimization pairs of problem oriented} = \frac{1}{2} \cdot \sum_{p=1}^{\mathcal{P}} C_{\sum_{u=1}^{\mathcal{U}} n_u}^2$$

It can be observed that when the number of users reaches 10, the number of problem-oriented optimization pairs increases by an order of magnitude compared to user-oriented optimization pairs. This is particularly advantageous for alleviating the data scarcity issue in code optimization domain.

## 2.2 MULTI-DIMENSION ANALYSIS

To rigorously and comprehensively compare code optimization pairs derived from two different perspectives, we employ a multi-faceted analysis. Specifically, based on the problem-oriented approach proposed in § 2.1, we reconstruct the PIE train pairs, resulting in the PCO (Problem-oriented Code Optimization). To ensure comparability and fairness, we retained the same number of optimization pairs for each problem in PCO as in the corresponding problem in PIE, selecting those with the top speedup rankings. This guarantees that both datasets contain a total of 78K optimization pairs, as shown in Table 5. We then perform comparative analyses across three different dimensions: *Structural Analysis*, *Semantic Representation Analysis*, and *Human & LLMs Sampling Analysis*.

**Structural Analysis.** First, we delve into the structural differences between "slow" and "fast" code within the optimization pairs. To achieve this, we utilize Control Flow Graphs (CFGs), which effectively capture the logical structure and execution pathways of a program. In order to quantify the structural differences, we employ the Graph Edit Distance (GED) metric. This metric measures the minimum edit operation cost between the CFGs of "slow" and "fast" code. As shown in Figure 2, significant differences emerge from different perspectives: user-oriented optimization pairs exhibit

a relatively small average GED, indicating that the optimizations involve minor changes, such as localized optimizations. In contrast, problem-oriented optimization pairs show a significantly higher average GED. This indicates that these optimizations often involve global changes, such as algorithmic adjustments and major structural modifications, which contrast sharply with the incremental nature of the user-oriented perspective. Detailed instances we shown in Figure 16 - 19.

**Semantic Representation Analysis.** Beyond examining the structural differences within optimization pairs, we further investigate the semantic differences between these pairs. Specifically, we concatenate the "slow" and "fast" code snippets within each pair. These concatenated sequences are subsequently encoded using the CODET5P-110M-EMBEDDING model (Wang et al., 2023), which generates semantic embeddings. To facilitate visualization, these embeddings are projected using t-SNE (van der Maaten & Hinton, 2008). As shown in Figure 3, the embeddings for user-oriented pairs are tightly clustered, indicating that the code pairs represent similar coding semantics. In contrast, the embeddings for the problem-oriented pairs are more dispersed, reflecting greater diversity.

**Human & LLMs Sampling Analysis.** Furthermore, we conduct a sampling analysis to investigate the optimization patterns. Specifically, we randomly select 100 pairs from the PIE and PCO for human analysis, aiming to classify the types of optimizations applied. The optimizations are categorized into three main types: global algorithmic optimizations, local optimizations, and other modifications (e.g., code cleanup), with details provided in the Appendix C. As shown in Figure 4, human analysis reveals distinct trends across the different perspectives: In PIE, true global algorithmic optimizations form a smaller share. In contrast, the majority of program pairs in PCO fall into

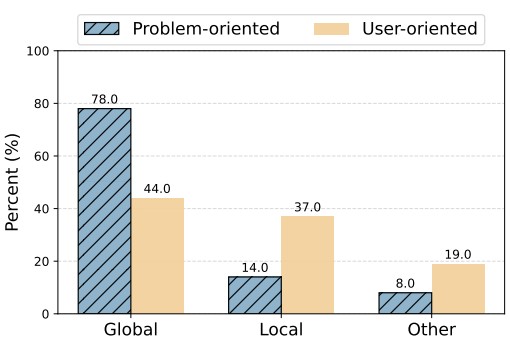

Figure 4: Human Analysis of the Optimization Types of Different Perspective.

the global algorithmic optimization category, indicating a stronger emphasis on significant algorithmic and structural improvements. The LLM analysis exhibits similar patterns, as shown in Figure 7.

## 2.3 ADAPTING LLMS TO OPTIMIZATION PAIRS

Subsequently, we conduct supervised fine-tuning to adapt LLMs to problem-oriented (PCO) and user-oriented (PIE) optimization pairs, to evaluate their performance in code optimization domain.

**Metrics.** To evaluate the optimization performance, we adopt the metrics from Shypula et al. (2024):

- **Percent Optimized** [%OPT]: The fraction of programs in the test set improved by a certain method. A program must be at least 10% faster and correct to contribute.
- **Speedup** [SPEEDUP]: The absolute improvement in running time. If $o$ and $n$ are the "old" and "new" running times, then $\text{SPEEDUP}(O, N) = \left(\frac{o}{n}\right)$. A program must be correct to contribute.
- **Percent Correct** [CORRECT]: The proportion of programs in the test set that are functionally equivalent to the original program (included as a secondary outcome).

We count a program as functionally correct only if it passes every test case. Additionally, we report SPEEDUP as the average speedup across all test set samples. For generated programs that are either incorrect or slower than the original, we use a speedup of $1.0\times$, hence, in the worst case, the original program has a speedup of 1.0 (further explanation is shown in Appendix G). We benchmark performance using the gem5 CPU simulator environment (Binkert et al., 2011) and compile all C++ programs with GCC version 9.4.0 and C++17 as well as the -O3 optimization flag. Therefore, any reported improvements would be those on top of the optimizing compiler.

**Code LLMs Selection.** We select GPT-4 (*0613*), GPT-4O (Achiam et al., 2023; OpenAI et al., 2024),CODELLAMA (Roziere et al., 2023), DEEPSEEK series (Guo et al., 2024; DeepSeek-AI, 2025) and QWEN2.5-CODER series (Hui et al., 2024) for code optimization. Detailed training configuration details, fine-tuning overhead, and GPU hours are provided in Appendix F.

**Decoding Strategy.** Code generation benefits from sampling multiple candidates and selecting the best one; in our case, the "best" refers to the fastest program that passes all test cases. We use

Table 1: Prompt and Fine-Tuning Results for LLMs on PIE and PCO with BEST@1 and BEST@8.

| Prompt / Dataset | LLMs & Code LLMs | BEST@1 | | | BEST@8 | | |
|---|---|---|---|---|---|---|---|
| | | %OPT | SPEEDUP | CORRECT | %OPT | SPEEDUP | CORRECT |
| Instruct | DEEPSEEKCODER 33B | 5.28% | 1.12× | 30.17% | 14.83% | 1.23× | 48.00% |
| Instruct | GPT-4 | 12.37% | 1.19× | 75.28% | 22.81% | 1.38× | **91.74%** |
| CoT | DEEPSEEKCODER 33B | 13.91% | 1.24× | 37.45% | 20.81% | 1.55× | 61.89% |
| CoT | GPT-4 | 23.43% | 1.37× | 48.65% | 47.92% | 1.74× | 80.53% |
| CoT | GPT-4O | 28.39% | 2.42× | 56.48% | 50.28% | 2.77× | 83.34% |
| CoT | DEEPSEEK-V3 | 31.92% | 2.78× | 58.93% | 53.86% | 3.02× | 87.32% |
| PIE | CODELLAMA 13B | 12.98% | 1.73× | 47.45% | 41.65% | 2.85× | 72.27% |
| PIE | DEEPSEEKCODER 7B | 23.56% | 2.29× | 41.27% | 47.23% | 3.34× | 69.23% |
| PIE | DEEPSEEKCODER 33B | 27.57% | 2.77× | 50.49% | 56.76% | 3.83× | 81.14% |
| PIE | QWEN2.5-CODER 7B | 26.96% | 2.80× | 41.21% | 56.17% | 3.85× | 78.54% |
| PIE | QWEN2.5-CODER 32B | 31.24% | 2.95× | 46.52% | 60.89% | 4.11× | 87.95% |
| **PCO** | CODELLAMA 13B | 31.83% | 3.23× | 44.26% | 55.87% | 4.89× | 69.61% |
| **PCO** | DEEPSEEKCODER 7B | 44.38% | 4.31× | 45.71% | 71.53% | 6.24× | 73.09% |
| **PCO** | DEEPSEEKCODER 33B | 49.83% | 4.57× | 50.64% | 74.87% | 6.67× | 78.29% |
| **PCO** | QWEN2.5-CODER 7B | 54.83% | 4.73× | 56.26% | 75.28% | 6.89× | 77.43% |
| **PCO** | QWEN2.5-CODER 32B | **58.90%** | **5.22×** | **61.55%** | **80.77%** | **7.22×** | 83.03% |

BEST@$k$ to denote this strategy, where $k$ represents the number of samples and the temperature is set to 0.7. we use vLLM (Kwon et al., 2023) for inference and detailed prompts are in Figure 10.

## 2.4 ADAPTING RESULTS.

**Instruction Prompting.** First, we use instruction prompts to guide the LLMs in optimizing code. Additionally, inspired by Chain-of-Thought (Wei et al., 2022), we ask the LLMs to reason about how to optimize the program before generating the optimized version. Details of Instruction/CoT prompts are in Appendix M. Table 1 shows that using instruct prompt and CoT did not significantly improve %OPT and SPEEDUP. The best performance by DEEPSEEK-V3 achieved 53.86%OPT and 3.02×SPEEDUP under BEST@8. Additionally, we observe that using CoT for optimization speeds up the program but can lead to a decline in CORRECT due to the complexities it introduces.

**Fine-Tuning Results.** As shown in Table 1, whether for different LLM series or varying parameter scales, significant performance differences are observed when finetuned on user-oriented (PIE) and problem-oriented (PCO) optimization pairs. QWEN2.5-CODER 32B on PCO at BEST@1, demonstrates substantial improvements: %OPT (31.24% → 58.90%), SPEEDUP (2.95×→5.22×), and CORRECT (46.52% → 61.55%) compared to finetuned on PIE. At BEST@8, %OPT and SPEEDUP reached 80.77% and 7.22×, respectively. This indicates a significant advantage in adapting to problem-oriented optimization pairs compared to user-oriented optimization pairs.

**Finding 1:** We observe that, unlike BEST@1, CORRECT slightly declines for most LLMs adapted on PCO under BEST@8 compared to PIE. This is because LLMs adapting on PCO results in more significant modifications to the code in pursuit of maximum efficiency, which slightly disrupts the balance of CORRECT. However, the gains in %OPT and SPEEDUP are substantial under BEST@8.

**Finding 2:** For LLMs adapted on PCO, both %OPT and CORRECT are much closer compared to PIE. This suggests that when the optimized code is correct, it is highly likely to be optimized. The closer %OPT and CORRECT are, the higher the proportion of "correct will be optimized". This insight also indicates that, for LLMs adapted on PCO, to further increase the optimization ratio and speedup, the performance bottleneck lies in ensuring correctness.

**PCO Percentage Analysis.** We further explore how fewer PCO optimization pairs impact %OPT, SPEEDUP, and CORRECT. To investigate this, we randomly selected a certain percentage of optimization pairs from PCO, reducing the number of pairs from 90% down to 10%, and fine-tuned QWEN2.5-CODER 32B in the same way. As shown in Figure 5, even with just 30% of the PCO optimization pairs, LLMs adapted on PCO achieve both %OPT and SPEEDUP that surpass those

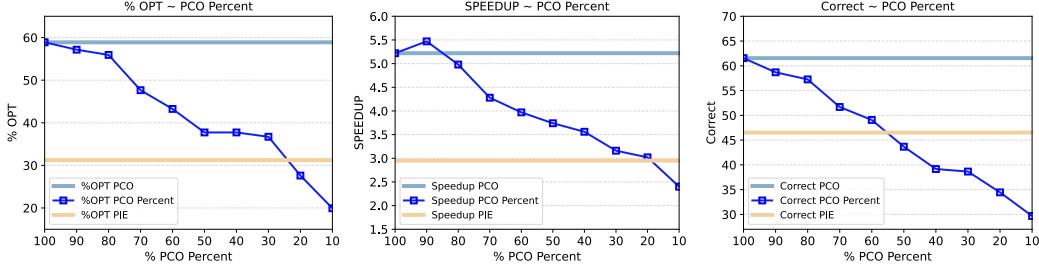

Figure 5: Performance impact of using varying percentages of PCO optimization pairs (from $100\% \rightarrow 10\%$) on %OPT, SPEEDUP, and CORRECT. The blue line represents the original PCO datasets, while the yellow line represents the original PIE datasets.

of the full PIE. Furthermore, with roughly half of the PCO pairs, CORRECT matches the full PIE. These results highlight the impressive data efficiency of the problem-oriented perspective, where fewer optimization pairs can still deliver competitive or even superior performance compared to full user-oriented optimization pairs.

**Disentangling Perspective and Selection Effects.** To rigorously evaluate the respective contributions of perspective versus optimization pairs selection strategy, we conducted a controlled ablation study by fine-tuning QWEN2.5-CODER 7B on three datasets under identical settings: PIE (original user-oriented), PCO-Random (randomly sampled from problem-oriented pairs while matching PIE's scale), and PCO-Top-Speedup (original PCO). Experimental details are provided in Appendix H. Results show PCO-Random substantially outperforms PIE across all metrics, confirming that the problem-oriented perspective itself drives performance gains. The additional improvement with PCO-Top-Speedup demonstrates that top-speedup sampling provides the best refinement. These findings establish that while selective sampling enhances performance, the core advantage stems from aggregating diverse solutions across users through the problem-oriented perspective.

**Learning Edit Patterns.** To further investigate whether PCO can distill effective algorithmic-improvement patterns from pairs with large structural disparities, we conducted an empirical study (Details in Appendix O). The study demonstrates that the edit-pattern (i.e., the algorithmic optimization pattern) learned by PCO is transferable and robust, rather than a mere conditional generation.

**Cost-Benefit Analysis of BEST@$k$ Strategy.** The BEST@$k$ strategy provides explicit control over the performance-computation trade-off in code optimization. Experimental results (as shown in Appendix K) demonstrate that while token costs scale linearly with the number of samples $k$, performance improvements follow a pattern of diminishing returns. This characteristic enables flexible deployment strategies: BEST@$1$ serves as the most efficient option for production environments with limited computational budgets, while progressively higher values of $k$ deliver enhanced optimization for resource-abundant scenarios. The strategy thus allows practitioners to balance optimization quality against inference costs according to their specific requirements.

**Validation on Physical Hardware.** We complement the gem5-based evaluation with performance measurements on physical hardware to assess real-world optimization effectiveness. Detailed results and analysis are provided in Appendix P.

## 3  ANCHOR VERIFICATION FRAMEWORK FOR PRACTICABILITY

In Section 2, we uncover the key challenge inherent in LLM code optimization. Whether through instruct prompting or finetuning, there is always a risk that optimized code may not be 100% correct. We refer to this phenomenon as the "optimization tax". To tackle the challenge of "optimization tax", we propose a novel anchor verification framework that leverages the original "slow code" as a gold-standard verification anchor. Unlike refinement in genenral code generation, which often relies on potentially error-prone synthetic test cases for refinement, the code optimization scenario has the unique advantage: the "slow code", despite its inefficiency, is functionally correct. This inherent characteristic positions it as an ideal test case verification anchor. Building on this insight, we design the anchor verification framework (Figure 6), which consists of three main stages:

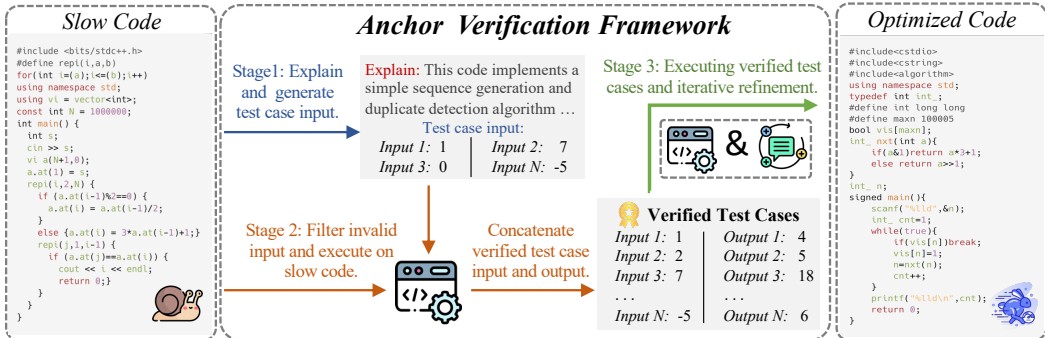

Figure 6: Anchor Verification Framework. It includes three stages: (1) generating test inputs based on the slow code's functionality, (2) constructing a verified test case set by executing inputs through the slow code, and (3) iteratively refining the optimized code with execution feedback.

Table 2: Results of Anchor Verification Framework and compared methods with QWEN2.5-CODER, GPT-4o, and DEEPSEEK-V3 on BEST@1. The baseline is the output of QWEN2.5-CODER 32B on **PCO** in Table 1. The improvement (denoted as $\Delta$) is measured against the baseline (w/o refinement).

| LLMs | Methods | BEST@1 | | | | | |
| --- | --- | --- | --- | --- | --- | --- | --- |
| | | %OPT | $\Delta \uparrow$ | SPEEDUP | $\Delta \uparrow$ | CORRECT | $\Delta \uparrow$ |
| | Baseline (w/o refinement) | 58.90% | | 5.22× | | 61.55% | |
| QWEN2.5-CODER 32B INSTRUCT | Self Debugging | 58.42% | -0.48 | 5.13× | -0.09 | 61.14% | -0.41 |
| | Direct Test Generation | 62.98% | +4.08 | 5.46× | +0.24 | 65.95% | +4.40 |
| | **Anchor Verification (Ours)** | **64.75%** | **+5.85** | **5.67×** | **+0.45** | **67.28%** | **+5.73** |
| GPT-4o | Self Debugging | 61.96% | +3.06 | 5.59× | +0.37 | 63.60% | +2.05 |
| | Direct Test Generation | 65.43% | +6.53 | 5.71× | +0.49 | 68.61% | +7.06 |
| | **Anchor Verification (Ours)** | **68.40%** | **+9.50** | **5.90×** | **+0.68** | **71.98%** | **+10.43** |
| DEEPSEEK-V3 | Self Debugging | 64.11% | +5.21 | 5.63× | +0.41 | 65.64% | +4.09 |
| | Direct Test Generation | 66.26% | +7.36 | 5.81× | +0.59 | 69.53% | +7.98 |
| | **Anchor Verification (Ours)** | **71.06%** | **+12.16** | **6.08×** | **+0.86** | **74.54%** | **+12.99** |

**Stage 1: Test Inputs Generation.** In the first stage, the LLM is prompted to explain the functionality of "slow code" and guided to generate a set of test inputs. These test inputs are designed to cover the boundary cases of the implemented functionality of "slow code". Unlike general LLM-based test case generation, this stage focuses solely on generating valid and meaningful test case inputs.

**Stage 2: Verified Testcase Construction.** Based on the obtained test case inputs in the first stage, we feed these inputs to the "slow code" for compilation and real execution. Although the "slow code" is inefficient, it can produce the correct execution results. We filter out test case inputs that don't match the input format and gather the corresponding output results. After that, we combine the test case inputs and corresponding outputs to form fully verified test case sets.

**Stage 3: Iterative Refinement.** Leveraging the verified test case sets, we compile and execute the optimized code to check its correctness. If any error occurs, similar to the feedback refinement mechanism in general code generation, we provide the execution error information to the LLM backbone, enabling it to iteratively refine the optimized code.

## 3.1  EXPERIMENT RESULTS.

**Compared Methods.** To rigorously validate the effectiveness of anchor verification framework, we benchmark against two compared methods. Details explanations of baselines are in Appendix Q.

- **Self-Debugging**: following Chen et al. (2024b), the self-debugging method prompts the LLM to provide line-by-line explanations of the generated code as a feedback signal for refinement.
- **Direct Test Generation**: the LLM directly generates complete test cases (including inputs and outputs) and uses these synthetic test cases to execute and iteratively refine the optimized code.

**Experiments Setup.** In the experiments, for all three methods, the maximum iteration count is set to 1. Detailed implementations and all corresponding prompts are also provided in Appendix Q.

**Main Results.** We use the output of "QWEN2.5-CODER 32B finetuned on PCO" as the baseline (the last row in Table 1). We experimented with three different LLM backbones: QWEN2.5-CODER 32B INSTRUCT, GPT-4o, and DEEPSEEK-V3, with the results shown in Table 2. All methods showed performance gains, except for a slight decline in the self-debugging with QWEN2.5-CODER 32B IN-STRUCT. The decline can be attributed to the high demands on the LLM's ability for self-explanation and correction, and QWEN2.5-CODER 32B INSTRUCT's overall performance still lags behind the other two LLMs. Anchor verification framework demonstrated the best improvements across all three LLM backbones, particularly with DEEPSEEK-V3. Compared to the baseline, CORRECT improved by 12.99%, %OPT improved by 12.16%, and SPEEDUP increased to 6.08×. This result confirms that CORRECT is the performance bottleneck, and that improving CORRECT can simultaneously enhance both %OPT and SPEEDUP. Additionally, we also performed experiments using the output of "QWEN2.5-CODER 32B finetuned on PIE" as the baseline, as shown in Table 14. The conclusions are similar, and anchor verification continues to deliver the highest performance gains.

**Root Cause of Performance Differences.** To further investigate whether the difference in test case output is the root cause of the performance difference, we conducted additional comparisons. In *Comparion Group*, the testcase outputs are generated by the LLM based on "slow code" and the testcase input, instead of real executing the "slow code". Everything else remained unchanged.

Table 3 shows that the *Comparison Group* falls short of the Anchor Verification. This indicates that the partially inaccurate outputs of test cases (approx.16%) do indeed negatively impact the subsequent refinement by the LLM and underscoring the necessity of executing test case inputs to obtain their outputs within Anchor Verification. Furthermore, *Comparison Group* performed slightly better than Direct Test Generation, suggesting that the two-step approach (LLM generating testcase inputs and outputs separately) places less burden on the LLM compared to directly generating the entire testcase. Consequently, the test case quality is superior.

Table 3: Root Cause of Performance Differences.

| DEEPSEEK-V3 | BEST@1 | | |
|---|---|---|---|
| | %OPT | SPEEDUP | CORRECT |
| Base | 58.90% | 5.22× | 61.55% |
| Direct Test Generation | 66.26% | 5.81× | 69.53% |
| Comparison Group | 68.30% | 5.91× | 70.86% |
| Anchor Verification | 71.06% | 6.08× | 74.54% |

**Increase the Number of Iterations.** In the experiment setup, the iteration count for all three methods was initially set to one iteration. To investigate the impact of multiple iterations on performance, we further verified the scenarios with three and five iterations. As shown in Table 4, the first iterations of both Direct Test Generation and Anchor Verification yield most significant performance gains. Moreover, for Direct Test Generation, the performance improvement from five iterations compared with three iterations is barely evident. In contrast, Anchor Verification still shows marked improvement. This further highlights the value of verified correct test cases for execution feedback refinement and alleviating the "optimization tax."

Table 4: Increase the Number of Iterations.

| DEEPSEEK-V3 | BEST@1 | | |
|---|---|---|---|
| | %OPT | SPEEDUP | CORRECT |
| Base | 58.90% | 5.22× | 61.55% |
| *Number of Iteration = 1* | | | |
| Direct Test Generation | 66.26% | 5.81× | 69.53% |
| Anchor Verification | 71.06% | 6.08× | 74.54% |
| *Number of Iteration = 3* | | | |
| Direct Test Generation | 68.30% | 5.89× | 70.76% |
| Anchor Verification | 74.85% | 6.19× | 77.81% |
| *Number of Iteration = 5* | | | |
| Direct Test Generation | 68.91% | 5.92× | 71.57% |
| Anchor Verification | 78.43% | 6.37× | 79.24% |

**Primary Reasons for Remaining Failures.** The empirical findings reveal an inherent trade-off between efficiency and correctness in program optimization: any attempt to accelerate a functionally correct but inefficient program risks introducing semantic deviations. Although the Anchor Verification Framework substantially reduces the error rate, it cannot eliminate persistent errors entirely. Consequently, achieving the ideal goal of "zero-correctness-loss optimization" remains an open challenge that deserves continued investigation. A systematic analysis of failures (shown in

Appendix I) reveals that roughly half fall into the "Compiled, but semantically wrong" category, indicating that current LLMs still have blind spots in comprehending high-level program intent.

**Practical Cost.** To investigate the overhead of the Anchor Verification Framework, we measured the corresponding time cost for each stage (as shown in Appendix L). The results reveal that its overhead is nearly identical to that of "Direct Test Generation". This similarity arises because the two approaches share all major stages (e.g., code comprehension); the only extra step is the local sandbox execution that runs in milliseconds and incurs minimal cost compared to other stages.

**Case Study.** Additionally, we present case studies to intuitively show specific examples and intermediate results of the Anchor Verification Framework, as illustrated in Figure 20, 21, and 22.

**Related Work.** The detailed review of related work is provided in Appendix B.

## 4 GENERALIZABILITY

**Generalizability Beyond C/C++.** To assess the cross-language applicability of the problem-oriented perspective and Anchor Verification Framework, we replicated the core experiments on `Python`. Following the same dataset construction procedure as for C++, Python versions of PIE and PCO datasets were built from CodeNet. Since Python is not compatible with the gem5 simulator, we utilized `cProfile` for fine-grained runtime performance analysis. Experimental results consistently demonstrate the approach's effectiveness in Python. The problem-oriented perspective yields substantial improvements, with PCO-adapted models showing significant gains in optimization rate and speedup compared to user-oriented counterparts. Higher correctness rates in Python align with established findings that LLMs generally exhibit stronger comprehension of Python semantics, confirming that the problem-oriented perspective's advantages are not language-specific. Furthermore, the Anchor Verification Framework was further validated in Python using DEEPSEEK-V3 to refine PCO-finetuned QWEN2.5-CODER 32B outputs. The framework consistently achieved best performance while significantly enhancing output correctness, reinforcing that leveraging functionally correct but inefficient code as a trusted anchor represents a general and effective strategy for optimization across programming languages. Detailed experimental results are provided in Appendix J.

**Discussion of Generalizability to Complex Real-World Scenarios.** While the problem-oriented perspective and the Anchor Verification Framework have demonstrated strong performance, their core design philosophy also exhibits potential for generalization to more complex real-world software optimization scenarios. In practical development environments, the problem-oriented perspective naturally extends to situations where multiple developers collaboratively optimize the same code component over extended periods. For example, in long-maintained open-source projects or large-scale enterprise codebases, sequential optimization attempts targeting specific performance bottlenecks—often documented in version control histories—can be viewed as problem-oriented optimization trajectories that the proposed approach can effectively leverage. In particular, performance-critical components typically undergo multiple optimization iterations by different developers, where each performance-improving commit represents a unique "solution" to the persistent optimization "problem" of that component. The sequence of such commits forms the exact type of problem-oriented optimization trajectory that this methodology is designed to capture and utilize. Similarly, the Anchor Verification Framework shows strong generalization potential through its component-level anchoring mechanism. In microservice architectures or modular systems, thoroughly tested yet suboptimal service modules can serve as reliable verification anchors. Extending this methodology to address system-level challenges and apply it to optimization problems in large-scale codebases represents a natural and important direction for future research.

## 5 CONCLUSION

In this paper, we introduce a problem-oriented perspective and an anchor verification framework for code optimization. The problem-oriented perspective not only enhances the diversity of optimization pairs but also significantly mitigates the data scarcity issue in the domain of code optimization. The anchor verification framework effectively alleviates the "optimization tax" while simultaneously elevating the optimization ratio, speedup, and correctness to new levels. We hope these insights will offer a practical and effective path toward advancing program efficiency.

ACKNOWLEDGMENTS

This work is supported by the Fundamental Research Funds for the Central Universities (Zhejiang University NGICS Platform).

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

## A  THE USE OF LARGE LANGUAGE MODELS.

In the preparation of this manuscript, LLMs are utilized as a general-purpose writing assistant. Its role was strictly limited to improving the grammar, clarity, and readability of the text. The LLMs are not used for research ideation, conducting experiments, or the generation of any core scientific content. The authors take full responsibility for all content presented in this paper, including any text revised with the assistance of the LLM.

## B  RELATED WORKS.

**LLMs for Code-Related Tasks.**  LLMs pre-trained on extensive code corpora have demonstrated remarkable capabilities in various programming tasks, including code completion, code generation, and code summarization (Li et al., 2022; Nijkamp et al., 2023; Roziere et al., 2023; Wei et al., 2023; Guo et al., 2024; Song et al., 2024; Wang et al., 2025). To enhance the accuracy of code generation, numerous techniques and frameworks have been proposed, such as execution feedback and self-correction mechanisms (Chen et al., 2024b; Zhong et al., 2024; Moon et al., 2024; Olausson et al., 2024). However, despite these advancements, the research of LLMs to code optimization, a field of both practical significance and considerable real-world challenges, remains underexplored in both academia and industry.

**Code Optimization.**  With Moore's law losing momentum, program optimization has become a central focus of software engineering over past few decades (Bacon et al., 1994; Kistler & Franz, 2003; Garg et al., 2022). However, achieving high-level optimizations, such as algorithmic changes, remains challenging due to the difficulty in comprehending code semantics. Previous research has employed machine learning to enhance performance by identifying compiler transformations (Bacon et al., 1994), optimizing GPU code (Liou et al., 2020), and automatically selecting algorithms (Kerschke et al., 2019). Recently, Shypula et al. (2024) introduced the first C/C++ dataset designed for program efficiency optimization, with preliminary results demonstrating the potential of LLMs in code optimization.

## C CATEGORIES OF OPTIMIZATION TYPES.

We categorize code optimization into three main categories: global algorithmic optimizations, local optimizations, and other optimizations.

- **Global Algorithmic Optimizations:** This type of optimization involves altering the algorithm itself to achieve significant performance improvements. Such changes can effectively reduce time complexity and enhance the speed of code execution. Examples include transforming recursive solutions into dynamic programming approaches, leveraging advanced mathematical theories, and restructuring complex data processing logic. These optimizations can lead to substantial gains in efficiency and scalability.
- **Local Optimizations:** These optimizations focus on improving specific parts of the code without changing the overall algorithm. They include enhancing I/O functions, optimizing read/write patterns to minimize runtime delays, and reducing computational complexity in certain sections of the code. By addressing these localized issues, programs can achieve more efficient execution and better resource utilization, ultimately leading to faster and more responsive applications.
- **Other Optimizations:** This category involves general code cleanup and refactoring aimed at improving code readability, maintainability, and overall quality. Examples include removing unnecessary initializations and redundant code, cleaning up outdated comments, and organizing the code structure more logically.

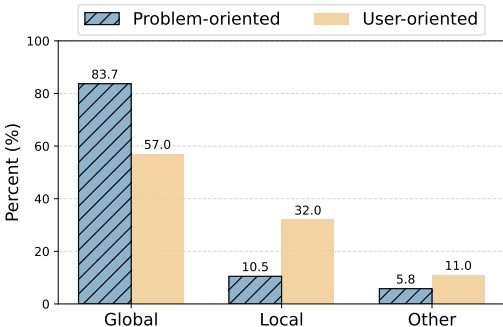

Figure 7: LLM Analysis of the Optimization Types between Problem-oriented and User-oriented Optimization Pairs.

## D LLMs ANALYSIS ON OPTIMIZATION TYPES.

Figure 7 presents the LLMs analysis of optimization types between problem-oriented and user-oriented optimization pairs. GPT-4 identifies a higher proportion of "global algorithm optimization" compared to human analysis. Upon further investigation, we find that this discrepancy is mainly due to GPT-4's tendency to categorize program pairs with significant changes as "global algorithm optimization".

## E DATASETS STATISTICS.

The statistical results of the PCO and PIE are shown in Table 5. We meticulously reviewed and ensured that any particular competitive programming problem appeared in only one of the train, validation, or test sets.

## F TRAINING DETAILS.

For Instruction/CoT prompt, we utilize the corresponding *chat* versions, while for fine-tuning, we employ the *base* versions of these LLMs.

Table 5: Number of unique problem ids and pairs.

| Dataset | Unique Problems | Pairs |
|---------|-----------------|-------|
| PIE | 1,474 | 77,967 |
| PCO | 1,474 | 77,967 |
| Val | 77 | 2,544 |
| Test | 41 | 978 |

We fine-tuned the CODELLAMA (13B), DEEPSEEKCODER (7B, 33B), and QWEN2.5-CODER (7B, 32B) models using LLAMA-FACTORY (Zheng et al., 2024) on a server equipped with $8\times$A100 GPUs (NVIDIA A100 80GB). During the fine-tuning process, we employed LoRA (Hu et al., 2022) (with lora_rank=8 and lora_target=*all*), and for both the PIE and PCO datasets, we trained the LLMs for only 2 epochs. The largest model, Qwen2.5-Coder-32B, required approximately 48 GPU hours (8 GPUs × 6 hours) to complete fine-tuning on the PCO/PIE dataset (78 k pairs). All experiments were conducted using AdamW (Loshchilov & Hutter, 2019) optimizer with an initial learning rate 5e-5.

## G  EXPLANATION OF SPEEDUP METRIC.

For the SPEEDUP metric, we adopted the same definition in PIE (Shypula et al., 2024) without modification. The rationale behind assigning $1.0\times$ to failures is that code optimization cannot guarantee 100% correctness, as previously mentioned. Therefore, if the optimized code produces incorrect results, users in practice would discard it and revert to the original version—effectively meaning no speedup was achieved. Hence, we assign a $1.0\times$ SPEEDUP for failures to reflect this scenario.

## H  DISENTANGLING PERSPECTIVE AND SELECTION EFFECTS.

To rigorously isolate the effect of the perspective (Problem-oriented vs. User-oriented) from the effect of selection (top-speedup vs. random), we have conducted a new controlled ablation study. We designed and trained on the following three datasets using the same QWEN2.5-CODER 7B and identical hyperparameters to ensure a fair comparison:

- **PIE (Original)**: Serves as our baseline, representing the user-oriented perspective (78K pairs).

- **PCO-Random (New Control)**: To isolate the pure effect of the perspective, for each problem, we randomly sampled the same number of pairs from the problem-oriented optimization pair pool as are present for that problem in the PIE dataset. This ensures that PCO-Random and PIE are perfectly matched in terms of the number of pairs and the specific problems covered, with the perspective being the only variable (78K pairs).

- **PCO-Top-Speedup (Original PCO)**: Our initially proposed method, which selects the pairs with the highest speedup for each problem to form the PCO dataset (78K pairs).

Table 6: Comparative Results of Different Optimization Perspectives and Selection Strategies.

| QWEN2.5-CODER 7B | BEST@1 | | |
|------------------|--------|---------|---------|
| | %OPT | SPEEDUP | CORRECT |
| PIE | 26.96% | 2.80× | 41.21% |
| PCO-Random | 49.55% | 4.56× | 53.23% |
| **PCO-Top-Speedup** | 54.83% | 4.73× | 56.26% |

From Table 6, the comparison between PCO-Random and PIE demonstrates clear advantages of the problem-oriented perspective. Even with random selection from the problem-oriented pool, the model achieves substantially higher performance: %OPT improves from 26.96% to 49.55%,

SPEEDUP increases from 2.80× to 4.56×, and CORRECT rises from 41.21% to 53.23%. These improvements decisively show that the problem-oriented perspective itself is the primary source of gains, while selection bias toward high-speedup pairs plays only a secondary role. The added value of the selection strategy is evident when comparing PCO-Top-Speedup with PCO-Random. The full PCO-Top-Speedup method yields further performance improvements, indicating that within the diverse landscape of the problem-oriented perspective, consciously selecting pairs with larger speedups can better unlock the model's potential. This represents a meaningful refinement, building upon the foundational gains achieved through the perspective shift.

## I    PRIMARY REASONS FOR REMAING FAILURES.

We conducted a systematic analysis of optimization failures. Among 100 optimized programs that failed the test suite, we identified three representative failure modes; the results are shown in Table 7.

- *Compilation failures (syntax or type errors).*
- *Successful compilation with I/O-format issues, which we could manually correct to pass the tests.*
- *Semantic errors, i.e., the optimized code compiles but behaves incorrectly.*

Table 7: Failure mode analysis.

| Failure Mode | Percent |
|---|---|
| *Failed to compile (syntax/type errors)* | 16% |
| *Compiled, semantic right but input/output format failure* | 31% |
| *Compiled, semantic wrong* | 53% |

Notably, Table 7 reveals that roughly half of the failures fall into the third category: the model failed to fully and accurately capture the original code's semantics during optimization. This suggests that current LLMs still have blind spots in understanding high-level program intent.

## J    GENERALIZABILITY BEYOND C/C++.

To evaluate the cross-language applicability of the problem-oriented perspective and the Anchor Verification Framework, we replicated the core experiments on Python. We constructed the PIE/PCO dataset (Python version) from CodeNet, following the same methodology as for C/C++. Since Python is incompatible with the gem5 simulator, we employed `cProfile` for fine-grained runtime analysis. The dataset contains 58,327 training pairs (PIE-Python, PCO-Python), 1,832 validation pairs, and 756 test pairs. The results are shown in Table 8:

Table 8: Fine-Tuning Results for LLMs on PIE and PCO (Python Version) with BEST@1.

| Python Version | LLMs | BEST@1 | | |
|---|---|---|---|---|
| | | %OPT | SPEEDUP | CORRECT |
| PIE | DEEPSEEKCODER 7B | 45.18% | 2.41× | 70.15% |
| PIE | QWEN2.5-CODER 7B | 48.35% | 2.65× | 72.80% |
| PIE | QWEN2.5-CODER 32B | 53.50% | 3.12× | 76.04% |
| **PCO** | DEEPSEEKCODER 7B | 63.26% | 4.05× | 79.41% |
| **PCO** | QWEN2.5-CODER 7B | 76.18% | 4.81× | 83.33% |
| **PCO** | QWEN2.5-CODER 32B | 80.05% | 5.35× | 89.27% |

The problem-oriented perspective yields substantial improvements in Python. With QWEN2.5-CODER 32B, %OPT improves from 53.50% to 80.05% and SPEEDUP increases from 3.12× to 5.35×. The higher correctness rates compared to C++ align with observations that LLMs typically demonstrate better comprehension of Python semantics. These results confirm that the benefits of

the problem-oriented perspective are language-agnostic. We further evaluated the Anchor Verification Framework using DEEPSEEK-V3 to refine outputs from "QWEN2.5-CODER 32B fine-tuned on PCO", the results are shown in Table 9.

Table 9: Result of Anchor Verification Framework and compared methods with DEEPSEEK-V3 on BEST@1 under the outputs from "QWEN2.5-CODER 32B fine-tuned on PCO (Python Version)".

| DEEPSEEK-V3 | BEST@1 | | |
|---|---|---|---|
| | %OPT | SPEEDUP | CORRECT |
| Base (w/o refinement) | 80.05% | 5.35× | 89.27% |
| Self Debugging | 81.57% | 5.43× | 90.32% |
| Direct Test Generation | 81.94% | 5.56× | 92.83% |
| **Anchor Verification** | **83.32%** | **5.78x** | **95.35%** |

The Anchor Verification Framework achieves the best performance, significantly enhancing correctness. This demonstrates that the core principle—using functionally correct but slow code as a trusted anchor for test case generation—represents a general and effective strategy for code optimization across programming languages.

## K   COST-BENEFIT ANALYSIS OF BEST@$k$ STRATEGY.

The BEST@$k$ strategy offers explicit control over the trade-off between performance and computational cost. As demonstrated in Table 10, detailed scaling results for QWEN2.5-CODER 32B on PCO reveal a clear pattern: while computational cost (measured in token usage) increases linearly with $k$, performance improvements exhibit diminishing returns.

Table 10: Performance–cost scaling for BEST@$k$.

| PCO on QWEN2.5-CODER 32B | %OPT | SPEEDUP | CORRECT |
|---|---|---|---|
| BEST@1 | 58.90% | 5.22× | 61.55% |
| BEST@2 | 66.41% | 5.94× | 67.95% |
| BEST@3 | 70.71% | 6.16× | 72.24% |
| BEST@4 | 73.57% | 6.45× | 74.90% |
| BEST@5 | 76.44% | 6.75× | 77.46% |
| BEST@6 | 78.58% | 6.96× | 79.81% |
| BEST@7 | 80.11% | 7.13× | 81.24% |
| BEST@8 | 80.77% | 7.22× | 83.03% |

This observation provides practical guidance for deploying the strategy in real-world scenarios:

- For cost-sensitive production environments, BEST@1 delivers substantial improvements with minimal inference cost, representing the most efficient operating point.

- When computational resources are abundant, increasing $k$ continues to enhance performance, though the marginal benefits decrease progressively. This allows practitioners to flexibly balance performance and cost according to specific requirements and constraints.

## L   PRACTICAL COST

To investigate the overhead of the Anchor Verification Framework, we measured the corresponding time cost for each stage. During the Anchor Verification Framework process, the overhead is almost identical to that of "Direct Test Generation" because the only difference is that Anchor Verification asks the LLM to produce only the test-case inputs, whereas Direct Test Generation requires the LLM to produce both inputs and outputs. All other stages—such as understanding and explaining

Table 11: Average runtime overhead per method.

| Time Cost | Query | Execution (testcase output) | Execution (testcase) | Refinement |
|---|---|---|---|---|
| Self-Debugging | 13.68 s | — | — | — |
| Direct Test Generation | 9.27 s | — | 0.23 s | 15.85 s |
| Anchor Verification | 7.24 s | 0.22 s | 0.22 s | 15.16 s |

the code—remain the same. Furthermore, we have broken down the entire pipeline into individual steps and measured the corresponding per-stage time cost, as shown in the Table 11 below:

From the measurement results, it can be seen that the primary time overhead for different methods is attributed to the invocation of the GPT-4o API, whereas the local sandbox environment for executing the code is relatively significantly faster in comparison. Meanwhile, the Anchor Verification Framework relies on the correctness of the slow code and has a high tolerance for its speed—as long as the code can produce the correct output within a finite time, the generated test cases are considered valid and can be used for subsequent refinement.

However, going a step further, for scenarios involving practically intolerable slow code execution, several intervention strategies remain available. Two practically viable approaches include: (1) Execution Time Threshold: enforcing a configurable time budget that automatically discards test inputs exceeding this limit to maintain practical verification overhead; (2) Strategic Test Input Selection: implementing intelligent sampling of LLM-generated inputs—such as prioritizing critical execution paths or boundary cases—to maximize verification value while minimizing execution cycles. These methods preserve the Anchor Verification Framework's core advantages while effectively handling substantially slow code.

## M  THE PROMPTS OF ADAPTING LLM ON OPTIMIZATION PAIRS.

In this section, we present the prompts for adapting the LLM to optimization pairs. The instruction prompt is shown in Figure 8, the CoT (Chain of Thought) prompt is shown in Figure 9, and the vLLM inference prompt is shown in Figure 10.

```
Given the program below,
↪   improve its performance:

### Program:
{slow_code}

### Optimized Version:
```

Figure 8: Instruct Prompt.

```
Given the program, generate
↪   an efficiency improvement
↪   strategy to enhance its
↪   performance.

### slower program:
{slow_code}

### strategy:
LLMs generated potential
↪   strategy.

### optimized version:
```

Figure 9: Chain-of-thought Prompting.

## N  FINE-TUNING ON GED-STRATIFIED SUBSETS

To investigate whether high-disparity code pairs hinder learning in the problem-oriented dataset, we conducted a controlled ablation study using Graph Edit Distance (GED) as the stratification criterion. Three specialized training subsets were constructed from the PCO dataset, each representing distinct characteristics of optimization pairs:

```
Given the program below,
↪  improve its performance:

### Program:
{slow_code}

### Optimized Version:
```

Figure 10: Inference Prompt.

```
...
int subarraySum(int* nums, int numsSize, int
    k) {
    int count = 0;
    for (int i = 0; i < numsSize; ++i) {
        int sum = 0;
        for (int j = i; j < numsSize; ++j) {
            sum += nums[j];
            if (sum == k) ++count;
        }
    }
...
```

(a) PCO: Slow Code.

```
...
static Node* new_node(int key, int val) {
    Node* n = (Node*)malloc(sizeof(Node));
    n->key = key; n->val = val; n->next =
        NULL;
    return n;
}
#define HASH_SIZE 200003
static Node* table[HASH_SIZE];

int subarraySum(int* nums, int numsSize, int
    k) {
    int count = 0, sum = 0;

    for (int i = 0; i < HASH_SIZE; ++i)
        table[i] = NULL;
    put(0, 1);                    //
        prefix_sum[0] = 1

    for (int i = 0; i < numsSize; ++i) {
        sum += nums[i];
        count += get(sum - k);
        put(sum, 1);
    }
...
```

(b) PCO: Optimized Code.

```
...
    for (int i = 0; i < n; ++i)
        for (int j = 0; j < m; ++j)
            scanf("%d", &a[i][j]);

    while (q--) {
        int x1, y1, x2, y2;
        scanf("%d %d %d %d", &x1, &y1,
            &x2, &y2);
        --x1; --y1; --x2; --y2;   //
            0-index
        long long sum = 0;
        for (int i = x1; i <= x2; ++i)
            for (int j = y1; j <= y2; ++j)
                sum += a[i][j];
        printf("%lld\n", sum);
    }
...
```

(c) Practical Slow Code.

```
...
    for (int i = 0; i < n; ++i)
        for (int j = 0; j < m; ++j)
            scanf("%d", &a[i][j]);

    long long pre[1005][1005] = {0};
    for (int i = 1; i <= n; ++i)
        for (int j = 1; j <= m; ++j)
            pre[i][j] = pre[i-1][j] +
                pre[i][j-1] -
                pre[i-1][j-1] +
                a[i-1][j-1];

    while (q--) {
        int x1, y1, x2, y2;
        scanf("%d %d %d %d", &x1, &y1,
            &x2, &y2);
        long long sum = pre[x2][y2] -
            pre[x1-1][y2] - pre[x2][y1-1]
            + pre[x1-1][y1-1];
        printf("%lld\n", sum);
    }
...
```

(d) Practical Optimized Code.

Figure 11: Case of learning edit patterns.

- **PCO-High-GED**: For each problem, we selected the top 40% of optimization pairs with the highest Graph Edit Distance (GED), representing major global transformations.

- **PCO-Low-GED**: For each problem, we selected the top 40% of pairs with the lowest GED, representing minor, localized optimizations.

- **PCO-Random**: For each problem, we randomly selected 40% of pairs as a balanced baseline controlling for dataset size.

Table 12: Optimization Performance on GED-Stratified PCO Subsets.

| QWEN2.5-CODER 7B | BEST@1 | | |
| --- | --- | --- | --- |
| | %OPT | SPEEDUP | CORRECT |
| PCO-High-GED | 46.23% | 4.48× | 50.43% |
| PCO-Low-GED | 36.49% | 3.75× | 42.43% |
| PCO-Random | 40.32% | 4.11× | 46.94% |

We then fine-tuned separate QWEN2.5-CODER 7B models on each subset using identical hyperparameters and training procedures. The comparative results (Table 12) reveal two key findings:

1. High-GED pairs enhance rather than hinder learning: The model trained exclusively on high-GED pairs demonstrates superior performance across all metrics compared to both the low-GED model (+9.74% %OPT, +0.73× SPEEDUP) and the random subset model (+5.91% %OPT, +0.373× SPEEDUP). This clearly indicates that while differences in variable declarations and readability exist in high-GED pairs, they do not fundamentally confuse the model. Instead, the model successfully learns to focus on the underlying algorithmic improvements that drive performance gains.

2. Algorithmic innovation drives substantial performance gains: The consistent outperformance of the PCO-High-GED model reveals that learning fundamental algorithmic transformations provides greater value than learning incremental, localized improvements. The model demonstrates the capability to abstract away surface-level differences in favor of recognizing deeper algorithmic patterns.

## O  LEARNING EDIT PATTERNS

To further investigate whether the PCO approach can distill effective algorithmic-improvement patterns from pairs with large structural disparities, we conducted an empirical case study. As shown in Figure 11, within the PCO optimization pairs, a representative and efficient algorithmic paradigm is: *rewriting a nested double for `loop` into a pattern of "prefix-sum preprocessing + elimination of the inner loop"*, thereby removing the significant overhead introduced by the nested iteration.

(i) In the PCO method, a common algorithmic paradigm shift boils down to replacing the `naive double for loops` by a `prefix-sum + hash look-up` scheme. By utilizing the `prefix sum sum[0..I]` and a `hash table to record the occurrence counts` of historical prefix sums, reducing the overall time complexity to $\mathcal{O}(n)$.

(ii) In practice, **the same improvement pattern** is also applied: in matrix operations, a `2-D prefix-sum with constant-time queries` dramatically reduces the nested complexity of the original `multilevel loops` by first `precomputing a prefix sum matrix pre[i][j]`.

This demonstrates that the "edit" pattern (algorithmic optimization pattern) learned by PCO is transferable and robust, rather than merely conditional generation.

## P  PRACTICAL PERFORMANCE STUDY ON PHYSICAL HARDWARE

To further validate the optimization effectiveness beyond simulated environments, we conducted performance measurements on physical hardware. This study aims to verify whether the performance improvements observed in gem5 simulations translate to real-world systems.

Table 13: Performance Validation on Physical Hardware.

| Optimization Type | SPEEDUP (gem5) | SPEEDUP (Hardware) | Relative Error |
|---|---|---|---|
| Global (Dynamic Programming) | 8.5× | 7.1× | -16.5% |
| Global (Greedy) | 12.2× | 10.3× | -15.6% |
| Local (Loop Unrolling) | 1.8× | 1.5× | -16.7% |
| Global (Data Structure) | 5.1× | 6.2× | +21.6% |
| Local (Cache Optimization) | 3.3× | 2.8× | -15.2% |
| Global (Algorithm Change) | 15.4× | 12.8× | -16.9% |
| Local (Memory Access) | 2.1× | 1.9× | -9.5% |
| **Average** | **6.9×** | **6.1×** | **-11.6%** |

**Experimental Setup**. We executed test programs on a server with $2\times$Intel Xeon Platinum 8468 CPUs (48 cores/socket) featuring 192 MiB L2 cache and 210 MiB L3 cache, operating at 3.1 GHz. All programs were compiled with `GCC 9.4.0` and `O3` optimization flags, maintaining consistency with the gem5 experimental configuration.

**Methodology**. We randomly selected 20 optimization pairs from the test set, representing different optimization categories (including global algorithmic changes and local optimizations). For each pair, we measured average execution time over 10 runs and calculated speedup on physical hardware.

Comparative results demonstrate strong agreement between simulated and hardware measurements (Table 13). Key findings include:

1. Consistent Optimization Effectiveness: All code optimizations that demonstrated significant speedup in gem5 showed substantial performance improvements on physical hardware, confirming the real-world validity of the proposed optimization approach.

2. Strong Correlation: The Pearson correlation coefficient between gem5 and hardware speedups is 0.89, indicating that gem5 serves as an excellent predictor of relative performance trends despite architectural differences.

3. Systematic Performance Difference: The slightly lower speedups observed on hardware (-11.6% on average) are expected, as gem5's timing model cannot fully capture all microarchitectural features of modern Intel Xeon processors. However, this systematic difference does not affect the fundamental conclusion that the proposed optimization methods provide substantial performance benefits.

This hardware validation confirms that the reported performance improvements effectively translate to real-world systems, strengthening the practical significance of the optimization framework.

## Q  IMPLEMENTATION DETAILS OF THE ANCHOR VERIFICATION FRAMEWORK AND THE COMPARED METHODS.

- **Anchor Verification:** In the Anchor Verification Framework, for the test case inputs in Stage 1, we prompt the LLM to generate three test case inputs based on the "slow code", the detailed prompt as illustrated in Figure 12. In Stage 2 and Stage 3, for compiling and executing both the "slow code" and "optimized code", we compile all C++ programs using `GCC` version 9.4.0 with `C++17` and the `-O3` optimization flag. In Stage 3, we leverage the verified test case sets. If an error occurs, we provide the error information to the LLM, allowing it to iteratively refine the optimized code based on this feedback. The detailed prompt is shown in Figure 13.

- **Self-Debugging:** following the approach presented in Chen et al. (2024b), the method instructs the LLM to provide line-by-line explanations of the generated program as feedback, functioning akin to rubber duck debugging. In this process, the LLM is capable of autonomously identifying and rectifying bugs without requiring human intervention. The detailed prompt is shown in Figure 14.

- **Direct Test Generation:** The LLM generates complete test cases (including both inputs and outputs) and utilizes synthetic test cases to execute the optimized code, enabling iterative refinement.

The prompt for generating complete test cases is shown in Figure 15, while the iterative refinement prompt is the same as the one used in Stage 3 of the Anchor Verification, as depicted in Figure 13.

```
Given the program below,
↪    please explain and analyze
↪    its functionality, and
↪    provide 3 testcase inputs
↪    that fully consider
↪    boundary conditions and
↪    code coverage. Note that
↪    only the testcase inputs
↪    are required.

### Program:
{slow_code}

### Explanation:
{Your explanation here}

### Test case Inputs:
{Your testcase inputs}
```

Figure 12: Anchor Verification Framework Stage 1 (Test Inputs Generation) Prompt.

```
You are a code expert, and
↪    your task is to correct
↪    the functionally
↪    incorrect code based on
↪    test cases and execution
↪    feedback. Analyze the
↪    issues, apply the
↪    necessary fixes, and
↪    ensure the corrected code
↪    meets the expected
↪    functionality and pass
↪    the testcase.

### Incorrect Program:
{code}

### Explanation:
{explanation}

### Testcase:
{Testcase}

### Feedback from execution:
{Feedback}

### Your corrected code
↪    version:
```

Figure 13: Anchor Verification Framework Stage 3 (Iterative Refinement) Prompt.

# R    RESULTS OF ANCHOR VERIFICATION ON PIE.

We conducted experiments using "QWEN2.5-CODER 32B fine-tuned on PIE" as the baseline and compared it with other methods. The results, shown in Table 14, demonstrate that Anchor Verification consistently delivers the highest performance gains. On the DEEPSEEK-V3 backbone, we observed improvements in %OPT ($31.24\% \rightarrow 47.28\%$), SPEEDUP ($2.95\times \rightarrow 3.40\times$), and CORRECT ($46.52\% \rightarrow 65.32\%$). Furthermore, we found that the gains in optimization ratio and speedup brought by the Anchor Verification Framework's improvements in correctness on PIE were not as significant as those observed in PCO. For example, on the DEEPSEEK-V3 backbone, CORRECT increased by 18.8%, but SPEEDUP only improved by $0.45\times$. In contrast, on the PCO scenario, CORRECT increased by 12.99%, while SPEEDUP saw a larger improvement of $0.86\times$.

# S    LIMITATIONS.

This paper focuses on optimizing the time efficiency of given code, without considering other optimization directions. However, in actual practice, there is a wide range of optimization avenues, such as memory optimization. Moreover, ensuring the complete accuracy of code optimization is a multifaceted and intricate issue that deserves further exploration and research.

```
Below is a potentially
↪   problematic C++ program.
↪   Please provide a
↪   line-by-line explanation
↪   and correct any errors
↪   that may be present.

### Program:
{program}

### Explanation:
{Your explanation here}

### Revised Program:
{Your revised program here}
```

Figure 14: Self-Debugging Prompt.

```
Given the program below,
↪   please explain and
↪   analyze its
↪   functionality, and
↪   generate three
↪   comprehensive test cases
↪   that thoroughly cover
↪   boundary conditions and
↪   all code paths. Each
↪   testcase should include
↪   the input and the
↪   corresponding expected
↪   output.

### Program:
{slow_code}

### Explanation:
{Your explanation here}

### Test case:
{Your testcase}
```

Figure 15: The Prompt of Direct Test Generation Method.

Table 14: Results of Anchor Verification and compared methods with QWEN2.5-CODER, GPT-4o, and DEEPSEEK-V3 on BEST@1. The baseline is the output of QWEN2.5-CODER 32B on **PIE** in Table 1.

| LLMs | Methods | BEST@1 | | | | | |
|---|---|---|---|---|---|---|---|
| | | %OPT | Δ↑ | SPEEDUP | Δ↑ | CORRECT | Δ↑ |
| | Baseline (w/o refinement) | 31.24% | | 2.95× | | 46.52% | |
| QWEN2.5-CODER 32B INSTRUCT | Self Debugging | 35.69% | +4.45 | 3.02× | +0.07 | 53.74% | +7.22 |
| | Direct Test Generation | 38.74% | +7.50 | 3.08× | +0.13 | 57.49% | +10.97 |
| | **Anchor Verification (Ours)** | 40.48% | +9.24 | 3.17× | +0.22 | 59.09% | +12.57 |
| GPT-4o | Self Debugging | 37.47% | +6.23 | 3.06× | +0.11 | 55.65% | +9.13 |
| | Direct Test Generation | 39.64% | +8.40 | 3.13× | +0.18 | 57.86% | +11.34 |
| | **Anchor Verification (Ours)** | 42.50% | +11.26 | 3.32× | +0.37 | 63.60% | +17.08 |
| DEEPSEEK-V3 | Self Debugging | 40.61% | +9.37 | 3.23× | +0.28 | 59.62% | +13.10 |
| | Direct Test Generation | 40.17% | +8.93 | 3.18× | +0.23 | 58.73% | +12.21 |
| | **Anchor Verification (Ours)** | **47.28%** | **+16.04** | **3.40×** | **+0.45** | **65.32%** | **+18.80** |

# T   DETAILED EXAMPLES OF USER-ORIENTED AND PROBLEM-ORIENTED PERSPECTIVES.

We provide detailed examples, as shown in Figure 16, Figure 17, Figure 18, and Figure 19, to illustrate that in the original PIE, program optimization pairs are constructed through iterative submissions and optimizations by the same user for the same programming problem, which can be limited by the single programmer's thought patterns.

# U   CASE STUDY OF ANCHOR VERIFICATION FRAMEWORK.

We present three case studies to vividly illustrate specific examples of Anchor Verification, as depicted in Figure 20, Figure 21, and Figure 22. These cases offer a clear and intuitive understanding of how Anchor Verification Framework operates in practice.

```cpp
#include <iostream>
#include <stdio.h>
using namespace std;
typedef long ll;

int main() {
    int length;
    ll arr[200000];
    ll res[200000] = {0};
    ll temp = 0;
    ll m = 2147483647;
    scanf("%d", &length);
    for (int i = 0; i <
        length; ++i) {
        scanf("%ld",
            &arr[i]);
    }
    res[0] = arr[0];
    for (int i = 1; i <
        length; ++i) {
        res[i] += res[i - 1]
            + arr[i];
    }
    for (int i = 1; i <
        length; ++i) {
        temp = abs(res[length
            - 1] - res[i - 1]
            * 2);
        m = min(temp, m);
    }
    printf("%ld\n", m);
    return 0;
}
```

(a) user1, initialization version.

```cpp
#include <bits/stdc++.h>
using namespace std;

#define int long long
typedef vector<int> vi;

const int INF = 1e18 + 5;

void solve() {
    int n;
    cin >> n;
    vi v(n), pre(n);
    int mn = INF, s = 0;
    for(int i = 0; i < n;
        i++) cin >> v[i];
    pre[0] = v[0];
    for(int i = 1; i < n;
        i++) pre[i] = v[i] +
        pre[i - 1];
    for(int i = n - 1; i >=
        1; i--) {
        s += v[i];
        mn = min(mn,
            abs(pre[i - 1] -
            s));
    }
    cout << mn;
}

signed main() {
    speed;
    int t = 1;
    while(t--) solve();
}
```

(b) user1, iteration version.

```cpp
#include<cstdio>
const int MAX = 2e5 + 5;
int a[MAX];
int main() {
    int n;
    long long sum = 0;
    scanf("%d", &n);
    for (int i = 0; i < n;
        i++)
    {
        scanf("%d", a + i);
        sum += a[i];
    }
    long long
        left,right,temp;
    left = sum - a[n - 1];
    right = a[n - 1];
    long long min = left >
        right ? left - right
        : right - left;
    left = 0;
    for (int i = 0; i < n-2;
        i++) {
        left += a[i];
        right = sum - left;
        temp = left > right ?
            left - right :
            right - left;
        if (temp < min)
            min=temp;
    }
    printf("%d\\n", min);
    return 0;
}
```

(c) another user submitted version.

Figure 16: The three submitted code solutions all address problem "p03661", which asks for a split point in an array that minimizes the absolute difference between the sums of the two parts. Solutions (a) and (b) are different submissions from same user "u018679195". In (a), the prefix sum is calculated first, then the minimum difference is computed from start to finish. In (b), the prefix sum is also calculated first, but the minimum difference is computed from end to start, avoiding additional multiplication operations. Solution (c), from user "u353919145", calculates the difference between the left and right sums in real-time, requiring only one pass through the loop. It can be seen that solutions (a) and (b) only make local changes, while (c) constructs a more efficient algorithm.

```cpp
#include <bits/stdc++.h>

using namespace std;

#define int long long

const int N = 1e5 + 5, M = 5,
    inf = 1e15;

int dp[N][M], a[N];

char op[N];

int Sign(int x) {
    if (x % 2) return -1;
    return 1;
}

int32_t main() {
    for (int i = 0; i < N;
        i++) for (int j = 0;
        j < M; j++) dp[i][j]
        = -inf;
    int n; cin >> n >> a[0];
    for (int i = 1; i < n;
        i++) cin >> op[i] >>
        a[i];
    dp[0][0] = a[0];
    for (int i = 1; i < n;
        i++) for (int j = M -
        1; j >= 0; j--) {
        if (op[i] == '+')
            dp[i][j] = dp[i -
            1][j] + a[i] *
            Sign(j);
        else if (j) dp[i][j]
            = dp[i - 1][j -
            1] + a[i] *
            Sign(j);
        if (j + 1 < M)
            dp[i][j] =
            max(dp[i][j],
            dp[i][j + 1]);
    }
    cout << dp[n-1][0] <<
        "\n";
}
```

(a) user1, initialization version.

```cpp
#include <bits/stdc++.h>
using namespace std;

#define int long long
const int N = 1e5 + 5, M = 3,
    inf = 1e15;

int dp[N][M], a[N];
char op[N];

int Sign(int x) {
    if (x % 2) return -1;
    return 1;
}

int32_t main() {
    ios::sync_with_stdio(0),
        cin.tie(0),
        cout.tie(0),
        cout.tie(0);
    for (int i = 0; i < N;
        i++) for (int j = 0;
        j < M; j++) dp[i][j]
        = -inf;
    int n; cin >> n >> a[0];
    for (int i = 1; i < n;
        i++) cin >> op[i] >>
        a[i];
    dp[0][0] = a[0];
    for (int i = 1; i < n;
        i++) for (int j = M -
        1; j >= 0; j--) {
        if (op[i] == '+')
            dp[i][j] = dp[i -
            1][j] + a[i] *
            Sign(j);
        else if (j) dp[i][j]
            = dp[i - 1][j -
            1] + a[i] *
            Sign(j);
        if (j + 1 < M)
            dp[i][j] =
            max(dp[i][j],
            dp[i][j + 1]);
    }
    cout << dp[n-1][0] <<
        "\n";
}
```

(b) user1, iteration version.

```cpp
#include<cstdio>
#include<algorithm>
using namespace std;
const int MAXN=int(1e5+5);
typedef long long LL;
#define INF LL(1e15)
LL s1,s2,as,n;
LL sz[MAXN],fh[MAXN];
char c[5];
int main()
{
    scanf("%lld",&n);
    scanf("%lld",&as);
    getchar();
    for(LL i=1;i<=n-1;i++) {
        scanf("%s",c);
        scanf("%d",&sz[i]);
        fh[i]=c[0];
    }
    s1=s2=-INF;
    for(LL i=1;i<=n-1;i++) {
        if(fh[i]=='-') {
            as-=sz[i];
            s1-=sz[i];
            s2+=sz[i];
            s1=max(s1,s2);
            s2=max(as,s2);
        }
        else {
            as+=sz[i];
            s1+=sz[i];
            s2-=sz[i];
        }
        s2=max(s1,s2);
        as=max(s2,as);
    }
    printf("%lld",as);
}
```

(c) another user submitted version.

Figure 17: The above three code snippets all come from the problem "p03580", which involves maximizing the evaluated value of a given formula by adding an arbitrary number of pairs of parentheses and outputting the maximum possible value. (a) and (b) are from the same user "u1821171064", both employing dynamic programming algorithms with a time complexity of $\mathcal{O}(N * M)$, where N is the length of the sequence and M is the number of states. In (b), the number of states M is reduced, and input and output are optimized. (c) is from user "u863370423" and uses a greedy algorithm, which is suitable for problems with fewer current states where the global optimal solution can be achieved through local optimization, with a time complexity of $\mathcal{O}(N)$.

```cpp
#include <iostream>
#include <cstring>
using namespace std;
typedef long long LL;
#define F(i) for(int
    i=0;i<n;i++)

int d[555][555] = {0},
    c[555][555] = {0};

int qu(int l, int r) {
    if (l > r) return 0;
    if (d[l][r] != -1) return
        d[l][r];
    return d[l][r] = c[l][r]
        + qu(l + 1, r) +
        qu(l, r - 1) - qu(l +
        1, r - 1);
}

int main() {
    memset(d, -1, sizeof(d));
    int n, m, q;
    cin >> n >> m >> q;
    while (m--) {
        int l, r;
        cin >> l >> r;
        c[l][r]++;
    }
    while (q--) {
        int l, r;
        cin >> l >> r;
        cout << qu(l, r) <<
            endl;
    }
    return 0;
}
```

(a) user1, initialization version.

```cpp
#include <bits/stdc++.h>
using namespace std;

#define int long long
#define pb push_back
#define faster
    ios::sync_with_stdio(0)

const int N = 509;
vector<int> v[N + 5];

int32_t main() {
    faster;
    int n, p, q;
    cin >> n >> p >> q;
    int x, y;
    for (int i = 1; i <= p;
        i++) {
        cin >> x >> y;
        v[x].pb(y);
    }
    for (int i = 1; i <= n;
        i++) {
        sort(v[i].begin(),
            v[i].end());
    }
    while (q--) {
        cin >> x >> y;
        int ans = 0;
        for (int i = x; i <=
            y; i++) {
            ans +=
                upper_bound(
                v[i].begin(),
                v[i].end(),
                y)
                - v[i].begin();
        }
        cout << ans << "\n";
    }
    return 0;
}
```

(b) user1, iteration version.

```cpp
#include <cstdio>
#define int long long
#define dotimes(i, n) for
    (int i = 0; i < (n); i++)

using namespace std;

int rint() {
    int n;
    scanf("%lld", &n);
    return n;
}

void wint(int n) {
    printf("%lld\n", n);
}

signed main() {
    int N = rint();
    int M = rint();
    int Q = rint();
    int S[N + 1][N + 1];
    dotimes(R, N + 1)
        dotimes(L, N + 1)
            S[R][L] = 0;
    dotimes(i, M) {
        int L = rint();
        int R = rint();
        S[R][L]++;
    }
    dotimes(R, N)
        dotimes(L, N)
            S[R + 1][L + 1] += S[R
                + 1][L] + S[R][L +
                1] - S[R][L];
    dotimes(i, Q) {
        int p = rint() - 1;
        int q = rint();
        wint(S[q][q] + S[p][p] -
            S[q][p] - S[p][q]);
    }
    return 0;
}
```

(c) another user submitted version.

Figure 18: The above three code segments all come from the same problem "p03283", which deals with cumulative sum queries in a 2D matrix. (a) and (b) are different submission versions from the same user "u816631826". In (a), the problem is solved using recursion and dynamic programming, but the query time complexity is high, $\mathcal{O}\left(N^2\right)$. In (b), the STL-provided binary search function is used, reducing the time complexity to $\mathcal{O}\left(N * \log(N)\right)$. (c) comes from another user "u281670674" and solves the problem using a 2D prefix sum matrix. The preprocessing time complexity is $\mathcal{O}\left(N^2\right)$, but the query time complexity for each query is $\mathcal{O}\left(1\right)$, making it more efficient.

```cpp
#include <bits/stdc++.h>

using namespace std;

inline void rd(int &x) {
    char ch;
for(;!isdigit(ch=getchar()););
for(x=ch-'0';
isdigit(ch=getchar());)
    x=x*10+ch-'0';
}

typedef long long LL;

const int MAXN = 300005;

int N, n, a[MAXN], cnt[MAXN];

LL sum[MAXN];

int ans[MAXN];

inline bool chk(int k, int x)
    {
    int pos = upper_bound(a +
        1, a + n + 1, x) - a;
    return sum[pos-1] +
        1ll*(n-pos+1)*x >=
        1ll*k*x;
}

int main() {
    rd(N);
    for(int i = 1, x; i <= N;
        ++i) rd(x), ++cnt[x];
    for(int i = 1; i <=
        300000; ++i)
        if(cnt[i]) a[++n] =
        cnt[i];
    sort(a + 1, a + n + 1);
    for(int i = 1; i <= n;
        ++i) sum[i] =
        sum[i-1] + a[i];
    int now = 0;
    for(int k = n; k >= 1;
        --k) {
        while(now < N &&
            chk(k, now+1))
            ++now;
        ans[k] = now;
    }
    for(int i = 1; i <= N;
        ++i) printf("%d\n",
        ans[i]);
}
```

(a) user1, initialization version.

```cpp
#include <bits/stdc++.h>

using namespace std;

inline void rd(int &x) {
    char ch;
for(;!isdigit(ch=getchar()););
for(x=ch-'0';
    isdigit(ch=getchar());)
        x=x*10+ch-'0';
}

typedef long long LL;

const int MAXN = 300005;

int n, cnt[MAXN];

LL sum[MAXN];

int ans[MAXN];

inline bool chk(int k, int x)
    { return sum[x] >=
    1ll*k*x; }

int main() {
    rd(n);
    for(int i = 1, x; i <= n;
        ++i) rd(x), ++cnt[x],
        ++sum[cnt[x]];
    for(int i = 1; i <= n;
        ++i) sum[i] +=
        sum[i-1];
    int now = 0;
    for(int k = n; k >= 1;
        --k) {
        while(now < n &&
            chk(k, now+1))
            ++now;
        ans[k] = now;
    }
    for(int i = 1; i <= n;
        ++i) printf("%d\n",
        ans[i]);
}
```

(b) user1, iteration version.

```cpp
#include<bits/stdc++.h>
#include<cstdio>
using namespace std;
typedef long long ll;
#define rep(i, n) for(int i =
    0; i < (n); i++)
#define rep1(i, n) for(int i
    = 1; i <= (n); i++)

int hist[300002],
    cnt[300001];
const int cm = 1 << 17;
char cn[cm], * ci = cn + cm,
    ct;

inline int getint() {
    int A = 0;
    if (ci - cn + 16 > cm)
        while ((ct =
        getcha()) >= '0') A =
        A * 10 + ct - '0';
    else while ((ct = *ci++)
        >= '0') A = A * 10 +
        ct - '0';
    return A;}

const int dm = 1 << 21;
char dn[dm], * di = dn;

int main() {
    int N = getint();
    rep(i, N)
        hist[getint()]++;
    rep1(i, N)
        cnt[hist[i]]++;
    int k = 1;
    rep(i, N + 1) rep(j,
        cnt[i]) hist[k++] = i
    k = N + 1;
    int ruiseki = N;
    int mae = 0;
    for (int i = N; i >= 1;
        i--) {
        while (hist[k - 1] >=
            i) {
            ruiseki -=
                hist[--k];
        }
        int kei = N - k + 1 +
            ruiseki / i;

        for (int j = mae + 1;
            j <= kei; j++)
            putint(i);
        mae = kei;
    }
    for (int j = mae + 1; j
        <= N; j++) {
        *di++ = '0';
        *di++ = '\n';
    }
    fwrite(dn, 1, di - dn,
        stdout);
    return 0;
}
```

(c) another user submitted version.

Figure 19: The above three code snippets all come from the problem "p02890", which requires calculating, for each possible K value (from 1 to N), the maximum number of times K cards with different numbers can be selected and removed from N cards. (a) and (b) are from the same user "u990400947" and utilize prefix sum calculation and searching. The latter employs condition checking with a time complexity of $\mathcal{O}(N * \log(N))$. (c) uses a difference array, reducing the time complexity to $\mathcal{O}(N)$.

```
** Optimized but incorrect Code **

#include <stdio.h>
#include <algorithm>
using namespace std;
int n,k,i,a[100005];
int main(){
    scanf("%d%d",&n,&k);
    for(i=1;i<=n;i++) scanf("%d",&a[i]);
    sort(a+1,a+n+1);
    printf("%d\n",n-lower_bound(a+1,a+n+1,k)+1);
    return 0;
}
```

```
*Below is the Explanation of slow code and generated test case
↪    inputs*
**Explanation:**
This code counts how many elements in an array are greater than or
↪    equal to a given value k. Here's the breakdown:

Input: Reads n (array size) and k (threshold value), followed by an
↪    array h of size n.
Sorting: Sorts the array h in descending order.
Counting: Iterates through the sorted array and counts how many
↪    elements are greater than or equal to k.
Output: Prints the count.

**Test Inputs:**
Test case 1 input:
5 10
15 8 12 10 20

Test case 2 input:
4 5
1 2 3 4

Test case 3 input:
6 7
7 7 7 7 7 7
```

```
** Verified Test cases **
** Test Cases**
Test case 1 input:
5 10
15 8 12 10 20
Test case 1 output:
4

Test case 2 input:
4 5
1 2 3 4
Test case 2 output:
0

Test case 3 input:
6 7
7 7 7 7 7 7
Test case 3 output:
6
```

```
** Optimized and Correct
↪    Code**

#include <stdio.h>
#include <algorithm>
using namespace std;
int n, k, i, a[100005];
int main() {
    scanf("%d%d", &n, &k);
    for (i = 0; i < n; i++)
    ↪    scanf("%d", &a[i]);
    sort(a, a + n);
    printf("%d\n", n -
    ↪    (lower_bound(a, a + n,
    ↪    k) - a));
    return 0;
}
```

Figure 20: Case Study 1. The reason for the error is that the offset was not calculated correctly. The correct fix is to calculate the index using $lower\_bound(...) - a$.

```
** Optimized but incorrect Code **

#include <cstdio>
#include <algorithm>
using std::min;
#define MAXN 100003
int abs(int z){
    if (z<0) return -z;
    else return z;
}
int main(void){
    int N,K,i,j;
    int h[MAXN], p[MAXN];
    scanf("%d%d",&N,&K);
    for (i=0;i<N;i++)
    {
        scanf("%d",h+i);
    }
    p[N-1]=0;
    p[N-2]=abs(h[N-2]-h[N-1]);
    for(i=N-3;i>=0;i--)
    {
        p[i]=min(p[i+1]+abs(h[i]-h[i+1]),p[i+2]
        +abs(h[i]-h[i+2]));
    }
    printf("%d\n",p[0]);
    return 0;
}
```

```
*Below is the Explanation of slow code and generated test case
↪  inputs*
**Explanation:**
This code solves a dynamic programming problem to find the minimum
↪  cost to reach the n-th element from the first element in an
↪  array, where you can jump up to k steps back at each position.
↪  Here's the breakdown:

Input: Reads n (array size) and k (maximum jump distance), followed
↪  by the array num.
Initialization:
dp array is initialized to a large value, except dp[0] and dp[1],
↪  which are set to 0 (starting points with no cost).
DP Transition:
For each position i from 2 to n, it calculates the minimum cost to
↪  reach i by considering jumps from positions j (within [i-k,
↪  i-1]).
The cost is the absolute difference between num[i] and num[j],
↪  added to the cost to reach j.
Output: The minimum cost to reach the n-th element (dp[n]) is
↪  printed.

** Test Cases**
Test case 1 input:
5 2
1 3 5 2 4

Test case 2 input:
4 1
10 20 10 30

Test case 3 input:
6 3
1 2 3 4 5 6
```

```
** Verified Test cases **
** Test Cases**
Test case 1 input:
5 2
1 3 5 2 4
Test case 1 output:
5

Test case 2 input:
4 1
10 20 10 30
Test case 2 output:
40

Test case 3 input:
6 3
1 2 3 4 5 6
Test case 3 output:
5
```

```cpp
** Optimized and Correct Code**

#include <cstdio>
#include <algorithm>
#include <climits>

using std::min;
#define MAXN 100003
int abs(int z){
    if (z<0)
        return -z;
    else
        return z;
}

int main(void){
    int N,K,i,j;
    int h[MAXN], p[MAXN];
    scanf("%d%d",&N,&K);
    for (i=0;i<N;i++)
    {
        scanf("%d",h+i);
    }
    p[0] = 0;
    for(i=1;i<N;i++)
    {
        p[i] = INT_MAX;
        for(j=1;j<=K && i-j>=0;j++)
        {
            p[i] = min(p[i], p[i-j] + abs(h[i] - h[i-j]));
        }
    }
    printf("%d\n",p[N-1]);

    return 0;
}
```

Figure 21: Case Study 2. The reason for the error is that the code does not properly handle the jump limit $K$, and the state transition logic is incomplete, only considering the cost of adjacent positions and ignoring larger jump ranges. The correct modification is to use a double loop to dynamically update the minimum cost for each position by considering jumps up to $K$ steps backward, thus fully addressing all possible jump scenarios.

```
** Optimized but incorrect Code **

#include<cstdio>
#include<cstring>
#include<algorithm>
using namespace std;
int n,ans;
int a[10010];

int main(){
        scanf("%d",&n);
        for(int i=1;i<=n;i++) scanf("%d",&a[i]);
        while(1){
                int mx=a[1],id=1;
                for(int i=2;i<=n;i++)
                        if(a[i]>mx) mx=a[i],id=i;
                if(mx<n-1) break;
                ans+=mx/n;
                a[id]%=n;
                for(int i=1;i<=n;i++)
                        if(i!=id) a[i]+=mx/n;
        }
        printf("%d\n",ans);
}
```

```
*Below is the Explanation of slow code and generated test case
↪   inputs*

**Explanation:**
The code simulates a process where an array a of size n is
↪   repeatedly modified until the maximum value in the array is no
↪   greater than n-1. The process involves the following steps:

Input: The code reads the size of the array n and the elements of
↪   the array a.
Main Loop: The code enters an infinite loop that continues until
↪   the maximum value in the array is less than or equal to n-1. In
↪   each iteration:
The maximum value maxn in the array and its index id are
↪   identified.
If maxn is less than or equal to n-1, the loop terminates.
Otherwise:
The value at a[id] is updated to maxn % n (the remainder when maxn
↪   is divided by n).
All other elements in the array are incremented by maxn / n (the
↪   quotient when maxn is divided by n).
The operation count ans is incremented by maxn / n.
Output: The total number of operations ans is printed.

** Test Cases**
Test case 1 input:
2
0 7

Test case 2 input:
4
8 0 0 0

Test case 3 input:
4
1 8 3 4
```

```
** Verified Test cases **

** Test Cases**
Test case 1 input:
2
0 7
Test case 1 output:
6

Test case 2 input:
4
8 0 0 0
Test case 2 output:
2

Test case 3 input:
4
1 8 3 4
Test case 3 output:
9
```

```cpp
** Optimized and Correct Code**

#include<cstdio>
#include<cstring>
#include<algorithm>
using namespace std;

int n,ans;
int a[10010];

int main(){
    scanf("%d",&n);
    for(int i=1;i<=n;i++) scanf("%d",&a[i]);
    while(1){
        int mx=a[1],id=1;
        for(int i=2;i<=n;i++)
            if(a[i]>mx) mx=a[i],id=i;
        if(mx<n) break;
        ans+=mx/n;
        a[id]%=n;
        for(int i=1;i<=n;i++)
            if(i!=id) a[i]+=mx/n;
    }
    printf("%d\n",ans);
    return 0;
}
```

Figure 22: Case Study 3. The error is that the termination condition used $mx < n - 1$, which prematurely stopped the loop, while the correct condition is $mx < n$, ensuring the loop only stops when maximum value is less than $n$.

