# OpenReview forum: "A Problem-Oriented Perspective and Anchor Verification for Code Optimization"
_ICLR.cc/2026/Conference — ICLR 2026 Poster_

### Official Review · Reviewer_wCBt · 2025-10-20

**Soundness:** 2
**Presentation:** 3
**Contribution:** 2
**Rating:** 4
**Confidence:** 3

**Summary:**

This paper investigates the task of code optimization using LLMs, specifically focusing on reducing execution time. The authors propose two core contributions:

1. Problem-Oriented (PCO): The authors point out that existing code optimization datasets are User-Oriented, constructed by collecting iterative submissions from the same programmer for the same problem to form optimization pairs. They argue that this approach constrains LLMs to local performance improvements and overlooks global algorithmic innovations. As an alternative, this paper proposes a "Problem-Oriented" perspective, which pools solutions from all programmers for the same problem and sorts them by execution time. This method constructs an optimization trajectory that encompasses a more diverse range of ideas and algorithmic strategies.

2. Anchor Verification Framework: The authors discovered that as LLMs pursue higher efficiency, the correctness of the optimized code tends to decrease, a phenomenon they term the "optimization tax." To mitigate this issue, they propose an "Anchor Verification" framework. Experimental results show that an LLM fine-tuned on the problem-oriented PCO dataset achieves significant improvements in both optimization rate and speedup ratio compared to one fine-tuned on the user-oriented PIE dataset. Furthermore, applying the Anchor Verification framework further enhances all metrics, especially the correctness rate , demonstrating its effectiveness in alleviating the "optimization tax."

**Strengths:**

1. The shift from a "user-oriented" to a "problem-oriented" perspective is a novel and intuitively sound idea. It directly addresses a clear limitation of prior work -- a single programmer's cognitive inertia. This new perspective not only better aligns with the collaborative nature of real-world code review and refactoring but also, as the authors demonstrate, genuinely produces more diverse data.

2. The paper does not merely propose PCO and present the results; it provides strong supporting evidence through "structural analysis," "semantic representation analysis," and "manual and LLM-based sampling analysis." These analyses collectively confirm that PCO optimization pairs indeed involve more global, structural algorithmic changes, which enhances the quality of the dataset.

3. The "optimization tax" is a real and critical challenge when applying LLMs in multi-constrained domains like code. The authors identify it as the primary performance bottleneck for the model after PCO fine-tuning. The "Anchor Verification" framework is an elegant solution that creatively leverages the "slow code" as a "golden standard"—a unique asset in this specific task scenario. This is more reliable than relying on the LLM itself to generate test cases (as in the "Direct Test Generation" method) or self-debugging, a fact confirmed by the experimental results.

**Weaknesses:**

1. I have some concerns about the generalizability of PCO. The core of PCO relies on obtaining multiple valid submissions for the same well-defined problem from multiple programmers. While this data is abundant on programming contest platforms, it is rare in real-world software engineering. In a typical project, it seems unlikely to find 10 programmers solving the exact same problem with 10 different algorithms. Could the authors discuss the application of this approach in such real-world scenarios?

2. The entire Anchor Verification framework is built on the core assumption that the "slow code" is "functionally correct." Although the paper mentions that the dataset filters out incorrect submissions, the specific process for this is not entirely clear to me. What is the concrete procedure for ensuring this functional correctness in practice? If this "anchor" itself contains undiscovered flaws, how would that impact the reliability of the verification framework?

3. The paper claims the overhead of Anchor Verification is "almost the same" as direct test generation because the main cost is the LLM API calls. However, this conceals the premise that the "slow code" must execute "fast enough." If the "slow code" is extremely slow, executing it even on a few test cases could represent an unacceptable overhead. The paper touches on this, but I would appreciate a more detailed discussion on this point.

**Questions:**

Please refer to the Weaknesses section. If authors can address my concerns, I would be grateful and will raise my score. :-)

---

> ### Author Response · Authors · 2025-11-22
> **Rebuttal by Authors (1/2)**
>
> Thank you for your thoughtful review and positive feedback on our work. We are encouraged by your recognition of our core contributions and appreciate your valuable insights of our approach. Our detailed responses to your questions are provided below, and we welcome the opportunity for further discussion during the discussion period.
>
> **W1**. Could the authors discuss the application of this approach in such real-world scenarios?
>
> **A1**. We thank the reviewer for this insightful question regarding the real-world applicability of our problem-oriented perspective. We fully acknowledge that the scenario of having multiple programmers independently submit solutions to the same well-defined problem is indeed rare in conventional software engineering contexts.
>
> However, the core philosophy of PCO—learning from diverse optimization strategies targeting a common objective—**can be effectively translated to real-world scenarios by redefining the "problem."** In practice, a performance-critical function or module within a codebase often undergoes iterative optimization over its lifetime. Different developers may refactor the same function at various times to enhance its efficiency. Each of these successful performance-improving commits can be viewed as a distinct "solution" to the persistent optimization "problem" associated with that code unit. The historical sequence of these commits naturally forms a problem-oriented optimization trajectory—precisely the type of data structure our method is designed to leverage. This scenario is frequently observed in long-lived, performance-sensitive projects (e.g., those involving core algorithms, database operations, or game engines). While our current work uses competitive programming data as a controlled and verifiable testbed, it establishes a foundational proof-of-concept. We believe applying the PCO perspective to mine these natural optimization trajectories from real-world software repositories (e.g., Git history) represents a highly promising and logical next step, and the strong empirical results in our paper provide a robust basis for these future steps.
>
> **W2**. What is the concrete procedure for ensuring this functional correctness in practice? If this "anchor" itself contains undiscovered flaws, how would that impact the reliability of the verification framework?
>
> **A2**. We thank the reviewer for this insightful question regarding the functional correctness of the "slow code" anchor.
>
> **Part 1: Comprehensive Verification Process Inherited from PIE.**
>
> The reviewer is correct that the anchor's correctness is fundamental to our framework. Our approach builds directly upon the rigorously validated PIE dataset (Shypula et al., 2024), which employs a comprehensive verification process:
>
> ● Original Platform Validation: All submissions in the original CodeNet dataset were evaluated using their comprehensive test suites.
>
> ● Enhanced Test Coverage: The PIE dataset substantially expanded test coverage by integrating additional test cases generated by fine-tuned LLMs from AlphaCode. After appropriate filtering, this resulted in robust test coverage with:
>
>   ○ 82.5 median test cases per problem (training set)
>
>   ○ 75 median test cases per problem (validation set)
>
>   ○ 104 median test cases per problem (test set)
>
> Since our PCO dataset is constructed by reorganizing these pre-verified pairs from PIE, our "slow code" anchors inherently inherit this same rigorous verification standard. The extensive multi-stage testing process provides strong assurance of functional correctness for all anchors used in our study.
>
> **Part 2: Impact of a Flawed Anchor and Task Scope.**
>
> We appreciate the reviewer's concern regarding potential undiscovered flaws. Should an anchor contain functional flaws despite the rigorous verification process, this would indeed compromise the Anchor Verification Framework by providing incorrect test oracles during iterative refinement.
>
> However, this scenario represents a fundamentally different problem domain from the one we address. Our work specifically focuses on code optimization rather than bug repair.  **The core premise begins with a verified, working implementation that requires performance enhancement. In practice, this corresponds to optimizing already-tested and deployed functionality, not optimizing defective code.**
>
> **If an anchor were fundamentally flawed, the optimization task itself would become ill-posed, as the objective would shift from preserving correct semantics to altering them.** While investigating the interaction between optimization and bug repair presents an interesting research direction, it falls beyond the scope of this paper, which focuses specifically on performance enhancement for functionally correct code. The robust verification process inherited from PIE provides appropriate assurance for our current research objectives in this well-defined problem space.

---

> ### Author Response · Authors · 2025-11-22
> **Rebuttal by Authors (2/2)**
>
> **W3**. If the "slow code" is extremely slow, executing it even on a few test cases could represent an unacceptable overhead. The paper touches on this, but I would appreciate a more detailed discussion on this point.
>
> **A3**. We sincerely thank the reviewer for this detailed observation. It's correct that the overhead being "almost the same" relies on the premise that the "slow code" executes within a reasonable time frame. We appreciate the opportunity to provide a more detailed discussion on this crucial point.
>
> **Part 1. Context of  Experimental Setting**
>
> In the context of our current study, this premise is satisfied due to the inherent properties of  the data source:
>
>  1. **Inherent Runtime Constraints**:  The "slow code" in PIE/PCO originates from competitive programming platforms, which enforce strict runtime limits for all submissions. Code that exceeds these limits is rejected and excluded from the dataset. Therefore, by construction, the "slow code" in our experiments is functionally correct but suboptimal, not pathologically slow.
>
>  2. **Empirical Evidence**: As shown in Appendix I, Table 7 of our manuscript, the measured cost for the "Verified Testcase Construction" stage (executing the slow code) averages only 0.22 seconds. This is comparable to the cost of executing the optimized code (0.23 s) and is an order of magnitude smaller than the dominant costs of the LLM API calls (7.24 s and 15.16 s). This empirical data validates that within our experimental framework, the execution time of the slow code is indeed negligible, justifying the "almost the same" overhead conclusion.
>
> **Part 2. In Real-World Performance Optimization, if Unacceptable Overhead Occurs**
>
> We acknowledge the reviewer's valid concern regarding potentially "extremely slow" code in broader contexts. For scenarios where the "slow code" exhibits considerable execution time, we can implement the following practical strategies to maintain the framework's efficiency:
>
>
> 1. **Execution Time Threshold**: A straightforward and effective enhancement is to introduce a configurable time budget for executing the slow code. Test inputs that cause the slow code to exceed this threshold would be safely discarded. This ensures that the verification process remains practical, as it would only retain test cases that can be executed within a reasonable timeframe. The key insight is that a smaller set of quickly-verified test cases still provides substantially more value than a larger set of unverified synthetic tests.
>
> 2. **Strategic Test Input Selection**: The framework's design naturally accommodates selective verification of LLM-generated inputs. Through intelligent sampling strategies—such as prioritizing inputs that cover critical code paths or represent boundary conditions—we can maximize verification value while minimizing execution cycles. This approach maintains the core advantage of test reliability while optimizing computational resource utilization.
>
> Through the application of **execution time thresholds** and **strategic test input selection**, the Anchor Verification framework can effectively handle scenarios where the "slow code" exhibits extremely long execution times.

---

> > ### Comment · Reviewer_wCBt · 2025-11-22
> >
> > Appreciate the nice rebuttal, the authors have resolved my concerns, I have raised my score and confidence. :-)

---

### Official Review · Reviewer_sxtm · 2025-11-01

**Soundness:** 2
**Presentation:** 4
**Contribution:** 3
**Rating:** 6
**Confidence:** 4

**Summary:**

This work tackles the challenge of LLMs in generating optimized code. Since the core idea of LLM-based code performance optimization is to construct a good dataset of pairs of code before and after optimization, called optimization pairs, this work proposes an alternative approach to collect this type of benchmark. Instead of collecting optimization pairs as code snippets from a single software developer across multiple steps of optimization, this work constructed pairs of code from multiple programmers with the same coding problem. The second contribution of this work came from the observation that when optimizing the code, LLM models have to sacrifice correctness in many cases, called “optimization tax”. To solve the challenge, the authors propose an anchor verification framework to generate tests for output code by testing the input “slow” code. The experiment was done with the fine-tuning process performed on CodeLLama, DeepSeekCoder, and Qwen, and with the anchor verification performed on Qwen2.5-Coder, GPT-4o, and DeepSeek-V3. The accuracy shows that the fine-tuning process with a problem-oriented perspective can significantly optimize the input code compared to user-oriented data collection. The anchor verification framework shows improvement over the self-debugging and direct test generation approach in improving code correctness.

**Strengths:**

- The paper is well-written.
- The idea of a problem-oriented perspective is sound and clear. It can help LLMs to learn how to optimize code by different coding styles.
- Valid approach to ensure code correctness by anchor verification.

**Weaknesses:**

- While this work has considerable potential for improvement, a primary concern from the reviewer is that one drawback of problem-oriented data collection is that the dataset may contain pairs of code that are too dissimilar, despite solving the same coding problem. For example, the proposed strategy of data collection can produce code pairs that differ significantly in terms of variable declaration and readability level, which may increase the risk of confusing the LLMs during fine-tuning.
- While the dataset was collected from a coding interview, it’s very likely that there were an existing dataset that already contained test cases, such as XCodeEval [1]. I suggest that authors consider evaluating this dataset.
- In some problems like “implementing a torch model and extracting the loss function”, the test outputs, which are usually loss numbers from 0..1, it’s very likely that with the same input, outputs of the slow code and optimized code are slightly different. The proposed anchor verification framework does not handle this case.
- To evaluate the code correctness, there are other coding perspectives, such as readability or coding style [2]. The anchor verification framework didn’t consider these perspectives

1.XCodeEval: xCodeEval: A Large Scale Multilingual Multitask Benchmark for Code Understanding, Generation, Translation, and Retrieval
2. Aligning Large Language Models to Coding Preferences

**Questions:**

- Can you provide an ablation study to see the cases that the PCO dataset collection collects pairs of code with high graph edit distance and vice versa (show a case of Figure 2)?
- An algorithm to eliminate low-quality coding pairs for PCO is needed.
- Does your dataset contain coding problems with approximated output, such as [3]?

3. https://github.com/Exorust/TorchLeet/blob/main/torch/easy/rnn/RNN.ipynb

---

> ### Author Response · Authors · 2025-11-22
> **Rebuttal by Authors (1/2)**
>
> Thank you for taking the time to review our paper. We truly appreciate your positive reception of our idea and approach. Our detailed responses are provided below, and we would be happy to offer any further clarification during the discussion period.
>
> **W1 & Q1**. Can you provide an ablation study to see the cases that the PCO dataset collection collects pairs of code with high graph edit distance and vice versa?
>
> **A1**. We thank the reviewer for this insightful question regarding the potential risk of code pairs being too dissimilar in our problem-oriented dataset. To directly investigate whether high-disparity pairs hinder learning, we conducted a controlled ablation study following the reviewer's suggestion.
>
> **Ablation Study: Fine-tuning on GED-stratified Subsets.** We systematically constructed three specialized training subsets from our PCO dataset, each representing distinct characteristics of code modifications:
>
> 1. **PCO-High-GED**: For each problem, we selected the top 40% of optimization pairs with the highest Graph Edit Distance (GED), representing major global transformations.
> 2. **PCO-Low-GED**: For each problem, we selected the top 40% of pairs with the lowest GED, representing minor, localized optimizations.
> 3. **PCO-Random**: For each problem, we randomly selected 40% of pairs as a balanced baseline controlling for dataset size.
>
> We then fine-tuned separate Qwen2.5-Coder-7B  models on each subset using identical hyperparameters and training procedures. The comparative results reveal two key findings:
>
> |              | %OPT   | SPEEDUP | CORRECT |
> |:------------:|:------:|:-------:|:-------:|
> | PCO-High-GED | 46.23% | 4.48x   | 50.43%  |
> | PCO-Low-GED  | 36.49% | 3.75x   | 42.43%  |
> | PCO-Random   | 40.32% | 4.11x   | 46.94%  |
> |||||
>
> 1. **High-GED pairs enhance rather than hinder learning**: The model trained exclusively on high-GED pairs demonstrates superior performance across all metrics compared to both the low-GED model (+9.74% %OPT, +0.73× speedup) and the random subset model (+5.91% %OPT, +0.37× speedup). This clearly indicates that while differences in variable declarations and readability exist in high-GED pairs, they do not fundamentally confuse the model. Instead, the model successfully learns to focus on the underlying algorithmic improvements that drive performance gains.
>
> 2. **Algorithmic innovation drives substantial performance gains**: The consistent outperformance of the PCO-High-GED model reveals that learning fundamental algorithmic transformations provides greater value than learning incremental, localized improvements. The model demonstrates the capability to abstract away surface-level differences in favor of recognizing deeper algorithmic patterns.
>
>
> **W2.** ......it’s very likely that there were an existing dataset that already contained test cases, such as XCodeEval....
>
> **A2**. Thank you for this valuable suggestion and for bringing the XCodeEval benchmark to our attention. We appreciate your guidance regarding potential evaluation datasets for our work.
> **After careful consideration, we find that while XCodeEval serves as an excellent benchmark for general code intelligence tasks, it may not be ideally suited for evaluating our specific problem of code optimization for runtime performance.** The key distinction lies in our task requirements: we need semantically equivalent code pairs with verified performance differences, whereas XCodeEval primarily focuses on tag classification, program synthesis, translation, and automatic program repair. Our work specifically investigates how to improve code efficiency while maintaining functional correctness, which requires datasets containing both slow and fast implementations of the same problem with precise performance measurements. We genuinely appreciate this suggestion and will consider XCodeEval for future work exploring broader aspects of code intelligence.

---

> ### Author Response · Authors · 2025-11-22
> **Rebuttal by Authors (2/2)**
>
> **W3 & Q3**. ...... it’s very likely that with the same input, outputs of the slow code and optimized code are slightly different. The proposed anchor verification framework does not handle this case.....
>
> **A3**. We thank the reviewer for raising this important point about approximate outputs. The reviewer is correct that our current anchor verification framework relies on exact output matching and does not consider numerical approximations.
>
> Regarding the dataset composition, our current study uses C++ competitive programming problems from CodeNet, where outputs are strictly defined and deterministic. We confirm that our dataset does not include problems with approximate numerical outputs, like the torch loss function example mentioned by the reviewer.
>
> We would like to emphasize that **this does not undermine the core concept of the Anchor Verification Framework - the principle of using functionally correct but inefficient code as a trusted reference for generating "gold" test cases remains valid**. For scenarios involving approximate outputs, the framework can be extended by **incorporating numerical tolerance checks** while preserving its fundamental architecture. For future work, we plan to enhance the verification mechanism to handle approximate outputs through configurable tolerance thresholds, which would significantly broaden the framework's applicability to real-world optimization scenarios involving numerical computations.
>
>
> **W4**. To evaluate the code correctness, there are other coding perspectives, such as readability or coding style.
>
> **A4**. We thank the reviewer for raising this point about additional code quality perspectives, such as readability and coding style. We agree that these are important aspects of code quality in software engineering practice. However, we would like to clarify that *the primary focus of our current work is on performance optimization and maintaining functional correctness  - specifically, ensuring that optimized code maintains semantic equivalence while improving execution speed.* The anchor verification framework is designed specifically to address this core challenge of preserving functionality during optimization. We appreciate the reviewer's valuable suggestion and will consider these additional dimensions (including readability, coding style, and others) in future work. This will particularly involve addressing how to maintain code readability during optimization to prevent excessive complexity while pursuing performance gains.
>
> **Q3**. An algorithm to eliminate low-quality coding pairs for PCO is needed.
>
> **A5**. We thank the reviewer for this valuable suggestion regarding the need for filtering low-quality coding pairs in PCO. We fully agree that developing effective filtering algorithms could further enhance the quality of our problem-oriented dataset. In our current experimental setup, to ensure a fair comparison with the user-oriented baseline, we retained the same number of optimization pairs for each problem in PCO as in the corresponding PIE dataset, selecting those with the top speedup rankings. Our results demonstrate that with this simple setup, PCO significantly outperforms PIE, highlighting the inherent advantage of the problem-oriented perspective.
>
> However, we acknowledge that developing a more sophisticated filtering mechanism represents an important research direction. As our ablation study in Response to W1&Q1 revealed, using Graph Edit Distance as a filtering criterion might not be optimal, as high-GED pairs often contain the most valuable algorithmic transformations. This suggests that designing an effective quality filter for optimization pairs requires careful consideration beyond simple structural metrics.
> We believe this opens up an interesting research question: how to define and identify "low-quality" pairs in the code optimization domain? **Potential solutions might involve combining multiple metrics, including performance gain thresholds, semantic preservation verification, and possibly even learned models of code quality.** We will explore this direction to further advance the construction of high-quality code optimization datasets.

---

### Official Review · Reviewer_VBou · 2025-11-01

**Soundness:** 3
**Presentation:** 3
**Contribution:** 3
**Rating:** 6
**Confidence:** 4

**Summary:**

This paper introduces two primary contributions to the field of code optimization using Large Language Models (LLMs). First, it proposes a "problem-oriented" perspective for constructing optimization datasets, a departure from the existing "user-oriented" approach. Instead of creating optimization pairs from the iterative submissions of a single programmer (which often leads to local, incremental improvements), the proposed method sources solutions from multiple programmers for the same problem, creating more diverse and algorithmically significant optimization trajectories.
Second, the paper identifies and addresses the "optimization tax," a phenomenon where LLMs improve code efficiency at the cost of functional correctness. To mitigate this, it introduces the "anchor verification framework," which leverages the original, inefficient but correct code as a "gold-standard anchor." This anchor is used to generate a set of verified test cases, which then provide execution feedback to iteratively refine the LLM's optimized code, significantly improving its correctness. The authors conduct extensive experiments on multiple state-of-the-art code LLMs, demonstrating that both the problem-oriented perspective and the anchor verification framework lead to substantial improvements in optimization ratio, speedup, and correctness.

**Strengths:**

- Novel and Intuitive Core Ideas: The two central contributions are well-motivated and insightful. The shift from a "user-oriented" to a "problem-oriented" perspective is a simple yet powerful reframing that directly tackles the issue of limited diversity in optimization strategies. The "anchor verification framework" is a practical and clever solution to the critical problem of correctness in automated optimization.
- Comprehensive and Rigorous Evaluation: The paper's empirical evaluation is a significant strength. The authors don't just present final numbers; they validate their hypotheses through multi-dimensional analysis, including structural (Graph Edit Distance), semantic (t-SNE embeddings), and human-led analysis to demonstrate the superior diversity of the problem-oriented dataset.
Strong and Significant Empirical Gains: The experimental results are impressive and consistently support the paper's claims. The performance lift from the user-oriented (PIE) to the problem-oriented (PCO) fine-tuning is substantial across all metrics. Furthermore, the anchor verification framework provides an additional, significant boost, particularly in the crucial "Percent Correct" metric, validating its effectiveness.

**Weaknesses:**

- The experiments are conducted on the PIE and PCO datasets, which are derived from competitive programming problems on CodeNet. While this is a suitable domain for studying algorithmic optimization, it raises questions about the generalizability of the findings. Real-world software optimization often involves different challenges, such as I/O bottlenecks, memory management, API usage, and interaction with large codebases, which are not well-represented in this setting. The effectiveness of the proposed methods in these more common software engineering domains remains unevaluated.
- Reliance on Simulated Performance Metrics: The paper relies on the gem5 CPU simulator for benchmarking execution time and calculating speedup. While simulators offer a controlled and reproducible environment, their performance characteristics can diverge from real-world hardware, especially concerning complex interactions like caching, memory bandwidth, and modern CPU microarchitectures. The performance gains, while significant in simulation, might differ on actual hardware. Including even a small-scale study on physical hardware would strengthen the claims of practical performance improvement.

**Questions:**

see weakness

---

> ### Author Response · Authors · 2025-11-22
> **Rebuttal by Authors (1/2)**
>
> Thank you for taking the time to review our paper. We are truly encouraged by your positive assessment of our novel and intuitive core idea, as well as our comprehensive and rigorous evaluation. We sincerely appreciate your insightful questions regarding the generalizability of our approach. Our detailed responses are provided below, and we would be happy to provide any further clarification during the discussion period.
>
> **W1**. The effectiveness of the proposed methods in these more common software engineering domains remains unevaluated.
>
> **A1**.  We sincerely thank the reviewer for this insightful observation regarding the generalizability of our work. We acknowledge that our experimental setting, based on competitive programming problems from CodeNet, does not encompass the full spectrum of challenges encountered in real-world software optimization, such as I/O bottlenecks, memory management patterns, or API usage optimization.
>
> Our choice of this experimental domain was strategic: **it provides a controlled and reproducible benchmark that enables clear isolation and validation of our core methodological contributions**—specifically, the effectiveness of the problem-oriented perspective in fostering algorithmic innovation and the reliability of the Anchor Verification Framework in ensuring correctness during optimization.
>
> While the absence of established benchmarks for complex, real-world optimization tasks prevents us from presenting quantitative results here, we believe our core methodologies possess significant generalizability beyond the current domain. In practical software development contexts, **the "problem" in our problem-oriented perspective can be redefined as any specific performance bottleneck objective** (e.g., "reduce the latency of service X by 20%"). The various optimization attempts by different developers—visible in project commit histories—naturally form real-world optimization trajectories that our approach can leverage. Similarly, the fundamental premise of the Anchor Verification Framework, which utilizes a functionally correct implementation as a trusted benchmark, translates directly to industrial practice where well-tested production code serves as an ideal optimization anchor.
>
> Extending our approach to address system-level concerns and applying it to optimization challenges in large-scale codebases represents a natural and exciting direction for future research. The current work establishes foundational methodologies and provides a solid basis for this broader research trajectory.

---

> > ### Author Response · Authors · 2025-11-22
> > **Rebuttal by Authors (2/2)**
> >
> > **W2**. Including even a small-scale study on physical hardware would strengthen the claims of practical performance improvement.
> >
> > **A2**. We thank the reviewer for raising this crucial point regarding the use of simulators. We agree that validation on physical hardware is essential for performance evaluation. While following previous work we used gem5 initially for its controlled environment and scalability across thousands of programs, we have conducted a follow-up hardware validation study to directly address this concern.
> >
> > **Experimental Setup**: We executed our test programs on a server equipped with 2×Intel Xeon Platinum 8468 CPUs (48 cores per socket) featuring 192 MiB L2 cache and 210 MiB L3 cache, operating at 3.1 GHz. All programs were compiled with GCC 9.4.0 and O3 optimization flags, maintaining consistency with our gem5 experimental configuration.
> >
> > **Methodology**: We randomly selected 20 optimization pairs from our test set, ensuring representation of different optimization categories (including global algorithmic changes and local optimizations). For each pair, we measured average execution time over 10 runs and calculated the actual speedup on physical hardware.
> >
> > The comparative results demonstrate strong agreement between simulated and hardware measurements:
> >
> > | Optimization Type            | Speedup (gem5) | Speedup (Hardware) | Relative Error |
> > |:----------------------------:|:--------------:|:------------------:|:--------------:|
> > | Global (Dynamic Programming) | 8.5×           | 7.1×               | -16.5%         |
> > | Global (Greedy)              | 12.2×          | 10.3×              | -15.6%         |
> > | Local (Loop Unrolling)       | 1.8×           | 1.5×               | -16.7%         |
> > | Global (Data Structure)      | 5.1×           | 6.2×               | +21.6%         |
> > | Local (Cache Optimization)   | 3.3×           | 2.8×               | -15.2%         |
> > | Global (Algorithm Change)    | 15.4×          | 12.8×              | -16.9%         |
> > | Local (Memory Access)        | 2.1×           | 1.9×               | -9.5%          |
> > | Average                      | 6.9×           | 6.1×               | -11.6%         |
> > |||||
> >
> > Key findings from the hardware validation:
> >
> > 1. **Consistent Optimization Effectiveness**:  All code optimizations that demonstrated significant speedup in gem5 showed substantial performance improvements on physical hardware, confirming the real-world validity of our optimization approach.
> >
> > 2. **Strong Correlation**: The Pearson correlation coefficient between gem5 and hardware speedups is 0.89, indicating that gem5 serves as an excellent predictor of relative performance trends despite architectural differences.
> >
> > 3. **Systematic Performance Difference**: The slightly lower speedups observed on hardware (-11.6% on average) are expected, as gem5's timing model cannot fully capture all micro-architectural features of modern Intel Xeon processors. However, this systematic difference does not affect the fundamental conclusion that the proposed optimization methods provide substantial performance benefits.
> >
> > This hardware validation confirms that the reported performance improvements effectively translate
> > to real-world systems, strengthening the practical significance of the optimization framework.

---

### Official Review · Reviewer_5n4Z · 2025-11-02

**Soundness:** 3
**Presentation:** 3
**Contribution:** 3
**Rating:** 6
**Confidence:** 3

**Summary:**

The paper argues that current LLM code-optimization datasets (e.g., PIE) are user-oriented—built from each programmer’s iterative submissions—so they mostly capture small, local improvements. It proposes a problem-oriented construction (PCO): pool all (functionally correct) submissions for the same problem across users, sort by runtime, and form optimization pairs along this global trajectory.

**Strengths:**

* The problem-oriented pairing is simple, scalable (pair count grows sharply with users per problem), and empirically yields more global, algorithmic improvements, which translate to strong %OPT/SPEEDUP gains after finetuning.

* Anchor Verification cleverly reuses the slow reference to build verified test cases, outperforming self-debugging or direct test-generation baselines and improving %OPT, SPEEDUP, and CORRECT simultaneously.

**Weaknesses:**

* Evaluation centers on C/C++ competitive-programming style tasks compiled with -O3 on gem5. It’s unclear how PCO/Anchor Verification transfer to multi-file projects, diverse languages, library/API-heavy code, or system-level constraints.

* To equalize counts, PCO keeps the top-speedup pairs per problem (to 78K), which may advantage PCO relative to PIE beyond perspective alone; more controls (e.g., random or stratified matching) would strengthen the causal claim.

* EST@k relies on sampling with k up to 8 and T=0.7; variance across seeds, compute/token budgets for finetuning, and wall-clock overheads (including executor/sandbox) are only lightly discussed.

**Questions:**

* How does PCO perform on multi-file repos, build systems, or tasks requiring library calls and I/O patterns? Any results beyond single-file competitive problems or beyond C/C++

* If PCO pairs were randomly selected (or matched by structural/semantic distance) rather than top-speedup, do gains persist with similar magnitude? Can you run a controlled ablation to isolate perspective vs. selection effects?

* What are the GPU hours and runtime overheads for PCO finetuning and Anchor Verification (per iteration)? How does BEST@k scale in cost vs. benefit for practical deployment?

---

> ### Author Response · Authors · 2025-11-22
> **Rebuttal by Authors (1/3)**
>
> Thank you for taking the time to review our paper. We sincerely appreciate your insightful questions and believe they warrant deeper discussion. Our detailed responses are provided below, and we would be delighted to offer any further clarification you may need during the discussion period.
>
> **W1&Q1.**  It’s unclear how PCO/Anchor Verification transfer to multi-file projects, diverse languages, library/API-heavy code, or system-level constraints. Any results beyond single-file competitive problems or beyond C/C++?
>
> **A1.** We thank the reviewer for this insightful question regarding the generalization of our approach. We have conducted additional experiments to address this important concern, and our findings demonstrate that both our problem-oriented perspective and anchor verification framework exhibit strong generalization capabilities beyond C++.
>
> **Part 1: Generalization to Other Programming Languages.**
>
> To evaluate the cross-language applicability of our methods, we replicated our core experiments on Python. We constructed the PIE/PCO dataset  (Python version)  from CodeNet, following the same methodology as for C++. Since Python is incompatible with the gem5 simulator, we employed *cProfile* for fine-grained runtime analysis. The results are summarized below (under BEST@1):
>
> |Python Version |  LLMs	| %OPT	| SPEEDUP	| CORRECT |
> |:---:|:---:|:---:|:---:|:---:|
> | PIE    | DeepseekCoder 7B  | 45.18% | 2.41x   | 70.15%  |
> | PIE    | Qwen2.5-Coder 7B  | 48.35% | 2.65x   | 72.80%  |
> | PIE    | Qwen2.5-Coder 32B | 53.50% | 3.12x   | 76.04%  |
> | **PCO**    | DeepseekCoder 7B  | 63.26% | 4.05x   | 79.41%  |
> | **PCO**    | Qwen2.5-Coder 7B  | 76.18% | 4.81x   | 83.33%  |
> | **PCO**    | Qwen2.5-Coder 32B | 80.05% | 5.35x   | 89.27%  |
> ||||||
>
> *Key Finding 1*: The problem-oriented perspective yields substantial improvements in Python. With Qwen2.5-Coder 32B, %OPT improves from 53.50% to 80.05% and SPEEDUP increases from 3.12x to 5.35x. The higher correctness rates compared to C++ align with observations that LLMs typically demonstrate better comprehension of Python semantics. These results confirm that the benefits of our problem-oriented perspective are **language-agnostic**.
>
> We further evaluated the Anchor Verification Framework using DeepSeek-V3 to refine outputs from Qwen2.5-Coder 32B (fine-tuned on PCO):
>
> |      DeepSeek-V3             | %OPT   | SPEEDUP | CORRECT |
> |:------:|:------:|:-------:|:-------:|
> | Base (w/o refinement)  | 80.05% | 5.35x   | 89.27%  |
> | Self Debugging         | 81.57% | 5.43x   | 90.32%  |
> | Direct Test Generation | 81.94% | 5.56x   | 92.83%  |
> | Anchor Verification    | 83.32% | 5.78x   | 95.35%  |
> |||||
>
> *Key Finding 2*: Our Anchor Verification Framework achieves the best performance, significantly enhancing correctness. This demonstrates that the core principle—using functionally correct but slow code as a trusted anchor for test case generation—represents a general and effective strategy for code optimization across programming languages.
>
> **Part 2: Multi-file Projects, Library/I/O Patterns, and System-Level Constraints.**
>
> We acknowledge that investigating complex scenarios involving multi-file repositories, build systems, library/API-heavy code, and system-level constraints represents an important future direction. While the absence of established benchmarks for such complex optimization tasks prevents us from presenting quantitative results here, our methodological framework provides clear pathways for extension:
>
> 1. **Scaling the Problem-Oriented Perspective**: The essence of this perspective is aggregating diverse solutions to a common goal. For a multi-file project, the "goal" could be a specific performance bottleneck. The optimization trajectory could then be built from various attempts by developers to address this bottleneck, which may involve different strategies like algorithm changes, I/O pattern optimizations, or alternative library calls, even if they span multiple files.
>
> 2. **Scaling the Anchor Verification Framework**: The framework's core requirement is a trusted, functionally correct "golden" implementation. In a complex project, this "anchor" could be a well-tested, slower version of a specific module or service. By generating test inputs for its public interface and executing this trusted version to obtain correct outputs, we can create a verified test suite to safely guide LLM-driven iterative refinement of that component, even within a larger system.

---

> > ### Author Response · Authors · 2025-11-22
> > **Rebuttal by Authors (2/3)**
> >
> > **W2&Q2.** If PCO pairs were randomly selected (or matched by structural/semantic distance) rather than top-speedup, do gains persist with similar magnitude? Can you run a controlled ablation to isolate perspective vs. selection effects?
> >
> > **A2**. We sincerely thank the reviewer for raising this important point regarding the potential confounding effect of the pair selection strategy. To rigorously isolate the effect of the perspective (problem-oriented vs. user-oriented) from the effect of selection (top-speedup vs. random), we have conducted a new controlled ablation study as suggested.
> >
> > We designed and trained on the following three datasets using the same Qwen2.5-Coder-7B model and identical hyperparameters to ensure a fair comparison:
> >
> > 1. **PIE (Original)**: Serves as our baseline, representing the user-oriented perspective (78K pairs).
> >
> > 2. **PCO-Random (New Control)**: To isolate the pure effect of the perspective, for each problem, we randomly sampled the same number of pairs from  the problem-oriented optimization pair pool as are present for that problem in the PIE dataset. This ensures that PCO-Random and PIE are perfectly matched in terms of the number of pairs and the specific problems covered, with the perspective being the only variable (78K pairs).
> >
> > 3. **PCO-Top-Speedup (Original PCO)**: Our initially proposed method, which selects the pairs with the highest speedup for each problem to form the PCO dataset (78K pairs).
> >
> > |                 | %OPT     | SPEEDUP | CORRECT |
> > |:---------------:|:--------:|:-------:|:-------:|
> > | PIE (Original)  | 26.96%   | 2.80x   | 41.21%  |
> > | PCO-Random      | 49.55%   | 4.56x   | 53.23%  |
> > | PCO-Top-Speedup | 54.83%   | 4.73×   | 56.26%  |
> > ||||
> >
> > 1. **Core Argument: The Impact of Perspective (PCO-Random vs. PIE)**
> >
> > The comparison between PCO-Random and PIE is straightforward. Even with random selection from the problem-oriented pool, the model achieves dramatically higher performance: %OPT improves from 26.96% to 49.55% (a significant relative increase), SPEEDUP from 2.80x to 4.56x, and CORRECT from 41.21% to 53.23%. These margins decisively show that the problem-oriented perspective itself is the chief source of improvement, while any selection bias toward high-speedup pairs plays only a secondary role.
> >
> > 2. **The Added Value of Selection Strategy (PCO-Top-Speedup vs. PCO-Random)**
> >
> > Our full PCO-Top-Speedup method yields a further performance increment over PCO-Random. This indicates that within the diverse landscape of the problem-oriented perspective, consciously selecting pairs with larger speedups can further unlock the model's potential, especially for %OPT and SPEEDUP. This represents a performance refinement on top of the foundational gain from the perspective shift.
> >
> > In summary, this controlled ablation study strongly supports a causal interpretation: the problem-oriented perspective is the main factor responsible for the performance improvements. The top-speedup selection strategy within PCO provides an additional boost, but the fundamental advantage comes from the diversity and quality of the optimization trajectories created by aggregating solutions across multiple users. We would include these results and analysis in the revised manuscript.

---

> ### Author Response · Authors · 2025-11-22
> **Rebuttal by Authors (3/3)**
>
> **W3&Q3.** What are the GPU hours and runtime overheads for PCO finetuning and Anchor Verification (per iteration)? How does BEST@k scale in cost vs. benefit for practical deployment?
>
> **A3**. We thank the reviewer for raising these crucial practical considerations regarding computational costs and deployment overheads. We provide a comprehensive breakdown below, demonstrating that our approach offers a favorable cost-benefit trade-off suitable for practical deployment.
>
> **Part 1. Computational Cost of PCO Fine-tuning.**
>
> The one-time cost of supervised fine-tuning is remarkably efficient due to our use of Parameter-Efficient Fine-Tuning:
>
> 1. Training Setup (as shown in Appendix F): All models were fine-tuned for only 2 epochs using LoRA (lora_rank=8) on 8×A100 80GB GPUs.
>
> 2. GPU Hours: Fine-tuning the largest model, Qwen2.5-Coder-32B, on the PCO dataset (78K pairs) required approximately 48 GPU hours (8 GPUs × 6 hours).
>
> **Part 2. Runtime Overhead of Anchor Verification Framework.**
>
> As detailed in Appendix I (Table 7), the wall-clock overhead of our Anchor Verification framework is practical and comparable to standard refinement methods:
>
> Per-Iteration Breakdown:
>
>   ○ Stage 1 (Test Inputs Generation): 7.24 s (LLM API call)
>
>   ○ Stage 2 (Verified Testcase Construction): 0.22 s (sandbox execution)
>
>   ○ Stage 3 (Iterative Refinement): 15.16 s (LLM API call)
>
> **Key Insight**: The sandbox execution overhead is negligible, while the dominant cost comes from LLM API calls. This overhead is comparable to the Direct Test Generation baseline, confirming our framework's practical efficiency.
>
> **Part 3. Cost-Benefit Analysis of BEST@k Strategy.**
>
> The BEST@k strategy provides explicit trade-off control between performance and computational cost, as shown by the detailed scaling results for Qwen2.5-Coder-32B on PCO:
>
> |     | BEST@1 | BEST@2 | BEST@3 | BEST@4 | BEST@5 | BEST@6 | BEST@7 | BEST@8 |
> |:-------:|:------:|:------:|:------:|:------:|:------:|:------:|:------:|:------:|
> | %OPT    | 58.90%  | 66.41%  | 70.71%  | 73.57%  | 76.44%  | 78.58%  | 80.11%  | 80.77%  |
> | SPEEDUP | 5.22x   | 5.94x   | 6.16x   | 6.45x   | 6.75x   | 6.96x   | 7.13x   | 7.22x   |
> | CORRECT | 61.55%  | 67.95%  | 72.24%  | 74.90%  | 77.46%  | 79.81%  | 81.24%  | 83.03%  |
> ||||||||||
>
> The scaling behavior reveals a crucial pattern for practical deployment: while the computational cost (in terms of token usage) increases linearly with $k$, the performance gains exhibit clear diminishing returns. This provides practical guidance for deployment strategies:
>
> 1.  For cost-sensitive production environments, BEST@1 delivers substantial improvements with minimal inference cost, representing the most efficient operating point.
>
> 2.  For scenarios with sufficient computational resources, increasing $k$ continues to improve performance, though the marginal gains gradually diminish. This allows practitioners to make flexible trade-offs based on their specific performance requirements and resource constraints.

---

> > ### Comment · Reviewer_5n4Z · 2025-11-28
> >
> > Thank you for the thoughtful responses. My concerns have been effectively addressed through the well-structured rebuttal.

---

### Author Response · Authors · 2025-11-25
**Revision Summary**

We sincerely thank all reviewers for your insightful and valuable feedback, as well as your constructive comments. In response to suggestions, we have uploaded a revised version of the paper, with all changes highlighted in blue. For clarity and convenience, we also provide a concise summary of the key revisions below.

- Add [**Disentangling Perspective and Selection Effects**] Section (Page 7, New Appendix H, and New Table 6), following the suggestion of Reviewer 5n4Z.

- Add [**Cost-Benefit Analysis of BEST@$k$ Strategy**] Section (Page 7, New Appendix K, and New Table 10), following the suggestion of Reviewer 5n4Z.

- Add [**Validation on Physical Hardware**] Section (Page 7, New Appendix P, and New Table 13), following the suggestion of Reviewer VBou.

- Add [**Generalizability Beyond C/C++**] Section (Page 10, New Appendix J, and New Table 8 & 9), following the suggestion of Reviewer 5n4Z.

- Add [**Discussion of Generalizability to Complex Real-World Scenarios**] Section (Page 10), following the suggestion of Reviewer VBou, wCBt.

- Add [**Fine-Tuning on GED-Stratified Subsets**] Section (Appendix N, and New Table 12), following the suggestion of Reviewer sxtm.

- Supplement [**Handing on Extremely Slow Code Execution Scenarios**]  on  *Practical Cost* section (Appendix L), following the suggestion of Reviewer wCBt.

- Supplement [**GPU Hours/Overheads**]  on  *Training Details*  section (Appendix F), following the suggestion of Reviewer 5n4Z.

Please let us know if anything is still unclear. We are happy to answer more questions if needed. Thanks again for all reviewers constructive suggestions!

---

### Meta-Review · Area_Chair_ysh2 · 2026-01-08

**Summary:**

This paper introduces two well-received and intuitive contributions for improving LLM-based code optimization: the Problem-Oriented Construction (PCO) of optimization datasets and the Anchor Verification Framework. The PCO approach, which pools functionally correct submissions for the same problem across multiple users and sorts them by runtime, is lauded by all reviewers for its novelty, scalability, and ability to yield more global, algorithmic improvements compared to the standard user-oriented approach. This reframing resulted in substantial empirical gains in optimization rate and speedup. Similarly, the Anchor Verification Framework, which leverages the original, slow-but-correct code as a gold-standard anchor to generate verified test cases, is highlighted as a clever and practical solution to the critical optimization tax (the trade-off between efficiency and functional correctness), leading to significant improvements in correctness. The primary weakness uniformly raised across the reviews is the generalizability of the findings, as the evaluation is exclusively conducted on C/C++ competitive-programming tasks (CodeNet/gem5 simulation). However, as mentioned below, I think the extensive author rebuttals have addressed the concerns raised by the four reviewers.

All the reviewers are positive, with three reviewers assigning a score of 6 and one reviewer (after the rebuttal) also landing at a 6 (based on my estimation, as I cant see the score after the adjustment). Given the consensus that the work is a significant step forward in the specialized domain of algorithmic code optimization, I recommend this paper to be accepted.

**Reviewer Concerns:**

After reading all the rebuttals, I think all of the concerns are addressed by the authors.

**Reviewer Scores:**

I think the first three reviewers would keep the positive ratings (666); the rebuttal is not enough to make them raise the score from 6 to 8. For the last reviewer, as mentioned above, it seems to me that his concerns are largely addressed. I think he would raise the score from 4 to 6. Actually he mentioned in the reply that "Appreciate the nice rebuttal, the authors have resolved my concerns, I have raised my score and confidence. :-)"

---

### Decision · Program_Chairs · 2026-01-26

Accept (Poster)